# Simple within-stride changes in treadmill speed can drive selective changes in human gait symmetry

**Michael G. Browne**[1,2,3], **Jan Stenum**[1,2], **Purnima Padmanabhan**[1,4], **Ryan T. Roemmich**[1,2] *

**1** Center for Movement Studies, Kennedy Krieger Institute, Baltimore, MD, United States of America, **2** Dept of Physical Medicine and Rehabilitation, Johns Hopkins University School of Medicine, Baltimore, MD, United States of America, **3** Dept of Biomedical Engineering, University of Illinois at Chicago, Chicago, IL, United States of America, **4** Dept of Neuroscience, Johns Hopkins University School of Medicine, Baltimore, MD, United States of America

* rroemmi1@jhmi.edu

**Data Availability Statement:** Data are available at https://github.com/RyanRoemmich/DynamicTreadmill.

## Abstract

Millions of people walk with asymmetric gait patterns, highlighting a need for customizable rehabilitation approaches that can flexibly target different aspects of gait asymmetry. Here, we studied how simple within-stride changes in treadmill speed could drive selective changes in gait symmetry. In Experiment 1, healthy adults (n = 10) walked on an instrumented treadmill with and without a closed-loop controller engaged. This controller changed the treadmill speed to 1.50 or 0.75 m/s depending on whether the right or left leg generated propulsive ground reaction forces, respectively. Participants walked asymmetrically when the controller was engaged: the leg that accelerated during propulsion (right) showed smaller leading limb angles, larger trailing limb angles, and smaller propulsive forces than the leg that decelerated (left). In Experiment 2, healthy adults (n = 10) walked on the treadmill with and without an open-loop controller engaged. This controller changed the treadmill speed to 1.50 or 0.75 m/s at a prescribed time interval while a metronome guided participants to step at different time points relative to the speed change. Different patterns of gait asymmetry emerged depending on the timing of the speed change: step times, leading limb angles, and peak propulsion were asymmetric when the speed changed early in stance while step lengths, step times, and propulsion impulses were asymmetric when the speed changed later in stance. In sum, we show that simple manipulations of treadmill speed can drive selective changes in gait symmetry. Future work will explore the potential for this technique to restore gait symmetry in clinical populations.

## Introduction

Many clinical conditions–including stroke [1, 2], cerebral palsy [3], and lower limb amputation [4, 5]–result in gait asymmetry. Restoration of gait symmetry is a common goal of rehabilitation in these populations because asymmetric walking is associated with mobility

**Funding:** This project was funded by an American Heart Association Postdoctoral Fellowship (20POST35110071) to MGB, NIH grant R21 AG059184 to RTR and an American Heart Association Career Development Award to RTR (935556). The funders had no role in study design, data collection and analysis, decision to publish, or preparation of the manuscript.

**Competing interests:** The authors have declared that no competing interests exist.

impairment [6, 7] and higher risk of injury [8, 9]. Patients can exhibit asymmetry in many different aspects of the gait pattern, including kinematics (e.g., joint or limb angles [1, 10–15]), kinetics (e.g., ground reaction forces, joint moments or powers [16–20]), and spatiotemporal gait parameters (e.g., step lengths, stance times [1, 2, 6, 21]). Given the emerging foci on precision rehabilitation and development of patient-specific interventions, there is a need for new rehabilitation approaches that can be flexibly tailored to target various aspects of gait asymmetry.

Current rehabilitation strategies aim to restore gait symmetry using specialized treadmills [22–24], robotics [25–29], electrical stimulation [30–32], other technologies (e.g., biofeedback; [33–35]), and conventional gait training [36]. Several of these techniques have demonstrated encouraging results when implemented across multiple weeks of training [26, 31, 37, 38], though with important limitations. Because these techniques generally offer a targeted approach that aims to restore symmetry in a specific gait parameter (e.g., step length [37], propulsion [39, 40]), they are often best suited for select groups of patients with particular gait deficits. Furthermore, many of these approaches require specialized equipment that is not available in most clinics or rehabilitation centers.

Here, we aimed to address these limitations by developing an approach for driving selective changes in human gait symmetry that could be implemented using a single-belt (or, as in this study, tied-belt) treadmill. We also considered it important that our approach drive predictable changes in gait parameters that are common targets of rehabilitation in patients with gait asymmetry, including step length [37], propulsion [39, 40], and leading and trailing limb angles [41, 42]. To meet these criteria, we designed a paradigm that was inspired in part by prior studies of split-belt treadmill walking (i.e., walking on a treadmill with two independently controlled belts, one under each foot). Briefly, when healthy adults walk on a split-belt treadmill, many aspects of gait kinematics [22, 43] and kinetics [44, 45] become asymmetric when the treadmill belts move at different speeds simultaneously. Over time, people begin to step farther ahead (i.e., increase leading limb angle) onto the faster belt and allow the foot on the slower belt to extend further behind the body (i.e., increase trailing limb angle) [46]. This occurs with asymmetric changes in step lengths [43, 47] and a redistribution in the anteroposterior ground reaction forces produced by each leg [44]. In summary: when people walk in an asymmetric environment where the right leg steps onto/off of the treadmill at one speed and the left leg steps onto/off of the treadmill at a different speed, this drives predictable, asymmetric changes throughout the walking pattern.

In this study, we considered that a split-belt treadmill is not required to create this type of asymmetric walking environment. Rather, we hypothesized that we could create a situation where one leg steps onto (or off of) the treadmill while it moves faster (and the other leg does so while the treadmill moves slower) through simple changes in treadmill speed that occur at specific time points within the gait cycle. Moreover, we expected that this would drive asymmetric changes in kinematic, kinetic, and spatiotemporal gait parameters that depend on the timing of the speed change within the gait cycle.

We tested these hypotheses by performing two experiments. In Experiment 1, we used a closed-loop controller that changed the treadmill speed when each leg began to generate a propulsive ground reaction force. In Experiment 2, we tested an open-loop controller that changed the treadmill speed at prescribed time intervals coupled with a metronome to pace participant footfalls. We performed Experiment 2 to test whether we could implement this approach using a treadmill that provided no input to the controller (i.e., force data is not required to drive the speed changes), as this approach may be more amenable to clinical applications because it does not require an instrumented treadmill. We tested healthy adults to establish a scientific foundation about this approach before testing in patient populations.

## Materials and methods

### Participants

We tested two separate groups of healthy young adults in Experiments 1 and 2 (n = 10 in each experiment; Experiment 1: age (mean ± standard deviation): 22.3±3.6 years, 4M/6F; Experiment 2: age: 28.6±5.7 years, 4M/6F). All participants provided written, informed consent in accordance with the Johns Hopkins Medicine Institutional Review Board before participation. Participants reported no neurological, musculoskeletal, or cardiovascular conditions prior to enrollment. All participants received monetary compensation for their participation.

### Data collection

Participants performed all walking trials on an instrumented split-belt treadmill (Motek Medical, Amsterdam, Netherlands), though we emphasize that both belts moved at the same speed (i.e., "tied belts") during all trials. Participants wore a safety harness and did not hold onto the treadmill handrails. We collected kinematic data using a 10-camera three-dimensional motion capture system (Vicon Motion Systems, Centennial, CO; 100 Hz) with passive retroreflective markers placed on the seventh cervical vertebrae, jugular notch, tenth thoracic vertebrae, xiphoid process, second sacral vertebrae, and bilaterally over the iliac crest, anterior superior iliac spine, posterior superior iliac spine, trochanter, thigh (four-marker cluster), lateral femoral epicondyle, medial femoral epicondyle, shank (four-marker cluster), lateral malleolus, medial malleolus, calcaneus, first metatarsal head, second metatarsal head, and fifth metatarsal head. We collected triaxial ground reaction forces using separate force plates under each treadmill belt (1000 Hz).

### Experiment 1 protocol

The Experiment 1 protocol is shown in Fig 1A. Experiment 1 began with five two-minute trials of tied-belt walking at set speeds (i.e., *Baseline* trials: 0.75 m/s, 1.0 m/s, 1.25 m/s, 1.50 m/s, and again at 0.75 m/s to return the participants to slow walking prior to activating the closed-loop controller) to measure baseline reference data. Following the *Baseline* trials, we activated the closed-loop controller and participants walked on the treadmill for ten minutes (*Controller* trial). We developed a custom D-Flow (Motek Medical, Amsterdam, Netherlands) closed-loop treadmill controller that used anteroposterior ground reaction forces (AP GRF; 1000 Hz) to control the treadmill speed. The closed-loop controller set the treadmill to move at 1.50 m/s ("fast" speed) when the right AP GRF was positive (i.e., right leg generated a propulsive force) and 0.75 m/s ("slow" speed) when the left AP GRF was positive (treadmill acceleration set to 6.0 m/s$^2$). In other words, the treadmill speed changed back and forth between 1.50 m/s and 0.75 m/s and was controlled by the propulsive forces generated by each leg. Following the *Controller* trial, we deactivated the controller and participants walked at 0.75 m/s for ten minutes (*Washout* trial). We included the *Washout* trial to measure any aftereffects that may have resulted from the *Controller* trial.

### Experiment 2 protocol

The Experiment 2 protocol is shown in Fig 1B. Experiment 2 began with three five-minute trials of tied-belt walking at set speeds (i.e., *Baseline* trials: 0.75 m/s, 1.50 m/s, 1.125 m/s). These *Baseline* trials were five minutes long rather than two minutes as in Experiment 1 because we initially aimed to measure metabolic data during these trials (metabolic testing was removed from the protocol due to the COVID-19 pandemic). Custom MATLAB (MathWorks, Natick, MA) software first calculated the mean stride time for each participant across the 1.125 m/s

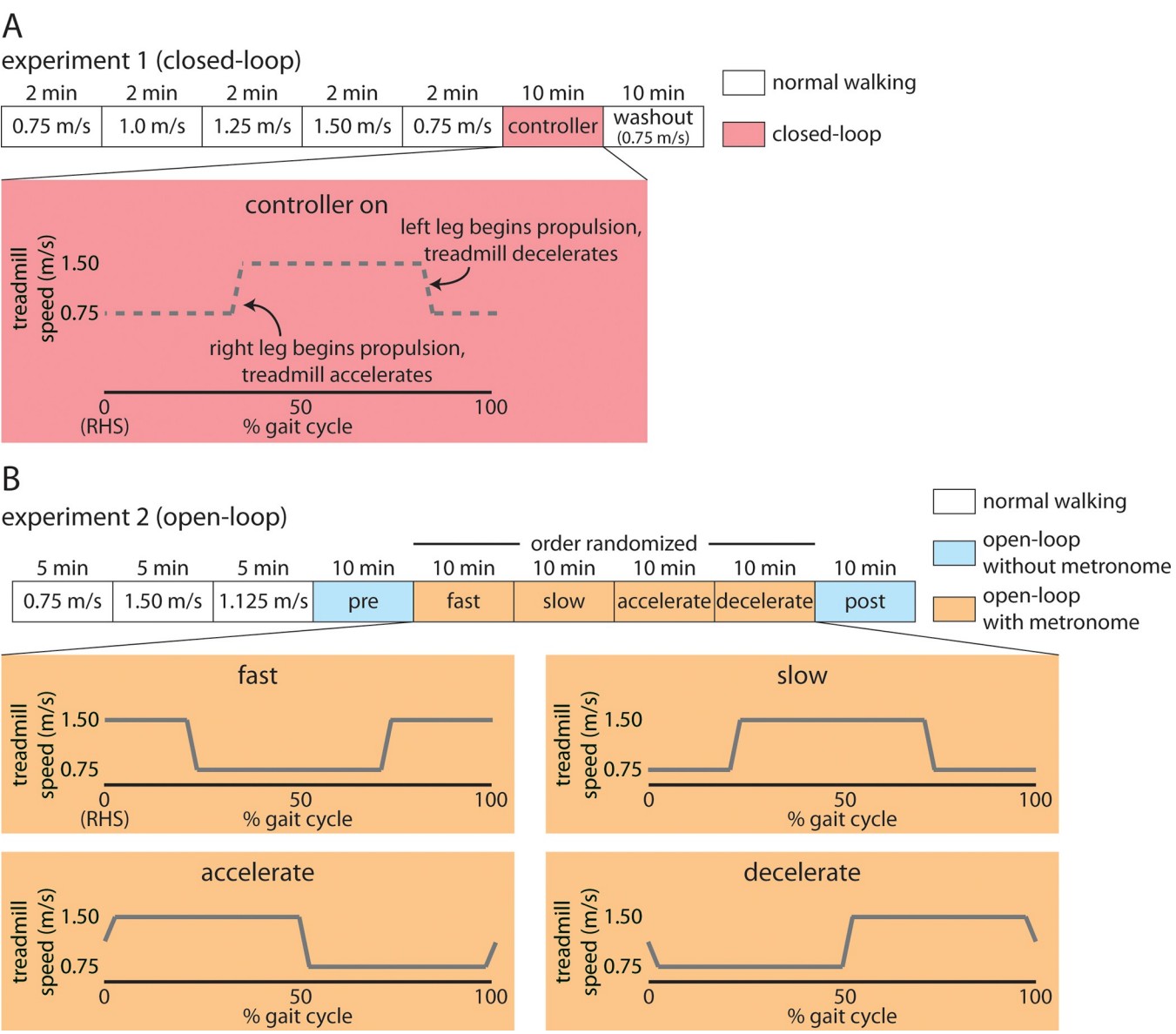

**Fig 1.** Experimental paradigms for A) Experiment 1, and B) Experiment 2. Both experiments began with baseline trials prior to testing different types of treadmill controllers. In Experiment 1, we used a closed-loop controller that changed the treadmill speed when the participant's right leg began to produce forward-directed propulsive forces. In Experiment 2, we used an open-loop controller where the experimenter set the changes in treadmill speed to occur at a fixed interval. We used a metronome to pace participants' footfalls at specific intervals in relation to the timing of the treadmill speed change.

*Baseline* trial by calculating the average of time differences between consecutive heel-strikes (based on a 20 N right vertical ground reaction force threshold) over one minute of walking. We used this mean stride time to program our open-loop controller. The open-loop controller consistently changed the treadmill speed from 0.75 m/s to 1.50 m/s and vice versa at time intervals that corresponded to 50% of the mean stride time during the 1.125 m/s *Baseline* trial. As an example, if the mean stride time for the 1.125 m/s *Baseline* trial was 1.2 seconds, the open-loop controller changed the treadmill speed every 0.6 seconds.

After the *Baseline* trials, participants walked with the open-loop controller activated for ten minutes without a metronome or other feedback (i.e., walking was unconstrained; *Pre* trial).

Following the *Pre* trial, participants performed four ten-minute walking trials (*Fast*, *Slow*, *Accelerate*, *and Decelerate*) in a randomized order with the open-loop controller activated and a metronome to pace their steps (*Metronome* trials). We synchronized the metronome beat with the treadmill speed changes to elicit the four different trials. Using the participant-specific stride time, the successive *Metronome* trials relied on custom D-Flow software generate four audible tones when the tied-belts were at 1.5 m/s (*Fast*), 0.75 m/s (*Slow*), 1.125 m/s while accelerating (*Accelerate*), and 1.125 m/s while decelerating (*Decelerate*). For each trial, we instructed participants to match their right heel-strike to the metronome for every stride across the ten-minute duration. Briefly, the during the Fast trial, the treadmill moved fast (i.e., 1.5 m/s) over the first and last ~25% of the gait cycle (i.e., right heel-strike to right heel-strike) and slow (i.e., 0.75 m/s) over the middle ~50% of the gait cycle; during the Slow trial, the treadmill moved slow over the first and last ~25% of the gait cycle (i.e., right heel-strike to right heel-strike) and fast over the middle ~50% of the gait cycle; during the Accelerate trial, the treadmill accelerated from slow to fast during heel-strike such that it would decelerate to slow in the last ~50% of the gait cycle; during the Decelerate trial, the treadmill decelerated from fast to slow during heel-strike such that it would accelerate to fast in the last ~50% of the gait cycle. This allowed us to observe how the participants' walking patterns changed when the treadmill speed changes occurred at different points in the gait cycle. Following the *Metronome* trials, participants again walked for ten minutes while the open-loop controller was activated but without a metronome (*Post* trial). We included this trial to assess whether participants developed a preference for a particular walking pattern after experiencing the four different ways of walking while the open-loop controller was engaged during the *Metronome* trials.

## Data analysis

We used custom MATLAB software for all data analysis. We filtered marker and ground reaction force data using fourth order low-pass Butterworth filters with cut-off frequencies of 6 and 28 Hz, respectively. We detected heel-strikes as the time points when the slope of the limb angle (calculated as a two-dimensional vector in the sagittal plane from the iliac crest to the second metatarsal head) trajectory changed from positive to negative (i.e., the limb began moving backward) and toe-offs as the opposite [48]. We calculated step length and step time as the difference between the lateral malleoli markers along the anteroposterior axis at heel-strike and time interval between consecutive heel-strikes, respectively. We calculated the leading limb angle as the limb angle at heel-strike and trailing limb angle as the limb angle at toe-off. We calculated peak propulsion and peak braking forces as the maximum and minimum values of the AP GRF for each leg within each gait cycle, respectively, and the propulsion and braking impulses by integrating the positive and negative sections of the AP GRF profile over time for each leg within each gait cycle, respectively. We calculated the asymmetry in each of these gait parameters using the following generalized equation: (right parameter−left parameter)/(right parameter+left parameter). We also recorded the belt speeds from the treadmill in Experiment 2 to confirm that they changed as desired with respect to the metronome pacing. In both Experiments 1 and 2, we defined the "early" epoch as the mean of the first 30 strides and the "late" epoch as the mean of the last 30 strides of each trial.

## Statistical analysis

After collecting data on four pilot participants, we performed a power analysis using G*Power [49]. Because propulsion was one of the key variables of interest in this study, we used propulsive impulse as the primary outcome measure in our power analysis and compared between the left and right legs during the *Controller* trial of Experiment 1. The power

analysis provided an estimated sample size of fewer than ten participants. To ensure robustness of our results (and to be similar to the sample sizes included in previous studies of locomotor learning and locomotor control), we included ten participants in each experiment as mentioned above.

In Experiment 1, we performed two sets of analyses. We designed the first analysis to assess how spatiotemporal, kinematic, and kinetic gait parameters changed between legs and over the course of the *Controller* trial as participants walked on the treadmill with the closed-loop controller engaged. We performed a 2 x 2 leg (left, right) x time (early, late) repeated measures ANOVA to compare step length, step time, leading limb angle, trailing limb angle, peak propulsive force, peak braking force, propulsive impulse, and braking impulse between legs and time epochs within the *Controller* trial. We also performed a series of one-sample t-tests to compare the asymmetries in each gait parameter against zero to confirm that all participants walked relatively symmetrically at baseline (the same analyses were also performed in Experiment 2). We designed the second analysis to assess aftereffects during the *Washout* trial. We performed a 2 x 2 leg (left, right) x time (second 0.75 m/s *Baseline* trial, early *Washout* trial) repeated measures ANOVA to compare the same battery of parameters listed above.

We also performed three sets of analyses in Experiment 2. We designed the first set of analyses to assess how spatiotemporal, kinematic, and kinetic gait parameters changed between legs and over the course of the open-loop controller trial in the different *Metronome* trials (*Fast*, *Slow*, *Accelerate*, and *Decelerate*). The trial names refer to the speed of the treadmill at right heel-strike (e.g., the treadmill moved fast at right heel-strike during the *Fast* trial). We performed a series of 2 x 2 leg (left, right) x time (early, late) repeated measures ANOVAs to compare the same parameters as in Experiment 1 between legs and time epochs within each of the four *Metronome* trials separately. Second, we performed 4 x 2 trial (*Fast*, *Slow*, *Accelerate*, *Decelerate*) x time (early, late) repeated measures ANOVAs to compare asymmetries for each of the gait parameters among the four *Metronome* trials and between time epochs. Finally, we designed the third analysis to assess whether participants developed a preference for a particular walking pattern from the *Pre* to *Post* trial (i.e., whether experiencing the different conditions during the *Metronome* trials led to a preference for a specific walking pattern when walking with the open-loop controller activated but no metronome constraint). We performed a 2 x 2 leg (left, right) x time (late *Pre* trial, late *Post* trial) repeated measures ANOVA to compare the same set of parameters. We set $\alpha \leq 0.05$, used IBM SPSS Statistics 25.0, and performed Bonferroni post hoc corrections for multiple comparisons where appropriate for all analyses in Experiments 1 and 2. To preserve the clarity of the text in the Results section, we report all statistical results in Tables 1–8.

## Results

### Experiment 1

The participants walked relatively symmetrically at baseline, as there were no significant differences in the asymmetry values for any gait parameters when compared to zero (all p > 0.05).

### *Controller* trial

When participants walked with the closed-loop controller activated during the *Controller* trial, we observed that they walked with significant asymmetries in step times (left larger than right; Fig 2), leading limb angles (left larger than right), and trailing limb angles (right larger–i.e., more negative–than left). Step lengths were not significantly asymmetric. Throughout the *Controller* trial, leading limb angles increased significantly over time in

**Table 1. Statistical results for Experiment 1 closed-loop *Controller* trial.**

| | Parameter | F(1,9) | p |
|---|---|---|---|
| **Factor: Leg** | | | |
| | Step length | 0.67 | 0.44 |
| | Step time | 42.17 | <0.001 |
| | Leading limb angle | 71.89 | <0.001 |
| | Trailing limb angle | 10.21 | 0.01 |
| | Peak propulsion | 14.41 | <0.01 |
| | Peak braking | 48.19 | <0.001 |
| | Propulsion impulse | 90.95 | <0.001 |
| | Braking impulse | 5.99 | 0.04 |
| **Factor: Time** | | | |
| | Step length | 2.96 | 0.12 |
| | Step time | 3.25 | 0.11 |
| | Leading limb angle | 9.16 | 0.01 |
| | Trailing limb angle | 0.65 | 0.44 |
| | Peak propulsion | 9.60 | 0.01 |
| | Peak braking | 0.01 | 0.94 |
| | Propulsion impulse | 10.61 | 0.01 |
| | Braking impulse | 8.56 | 0.02 |
| **Leg x Time Interaction** | | | |
| | Step length | 0.58 | 0.47 |
| | Step time | 0.93 | 0.36 |
| | Leading limb angle | 0.61 | 0.46 |
| | Trailing limb angle | 2.05 | 0.19 |
| | Peak propulsion | 3.57 | 0.09 |
| | Peak braking | 2.09 | 0.18 |
| | Propulsion impulse | 20.00 | <0.01 |
| | Braking impulse | 1.00 | 0.34 |
| | Post hoc Comparisons | | |
| | Propulsion impulse | | |
| | Early epoch, left vs. right | | <0.01 |
| | Late epoch, left vs. right | | <0.001 |
| | Left, early vs. late epoch | | <0.01 |
| | Right, early vs. late epoch | | 0.65 |

both legs. Participants showed significant asymmetries in peak propulsion (left larger than right; Fig 3), peak braking (left larger–i.e., more negative–than right), propulsion impulse (left larger than right), and braking impulse (right larger–i.e., more negative–than left). The horizontal dashed gray lines on all figures indicate mean values from each baseline trial (as labeled in Fig 2) to provide reference data. Full statistical results for the Experiment 1 *Controller* trial are shown in Table 1.

## Baseline and *Washout* trials

We also examined any potential aftereffects present during the *Washout* trial immediately following the *Controller* trial by comparing gait parameters during the *Washout* trial (gray shaded regions on Figs 2 and 3) to those observed during the final *Baseline* trial. We observed significant asymmetries in leading limb angles (left larger than right), propulsion impulse (left larger

**Table 2. Statistical results for Experiment 1 *Washout* vs. *Baseline* comparisons.**

| | Parameter | F(1,9) | p |
|---|---|---|---|
| **Factor: Leg** | | | |
| | Step length | 0.17 | 0.69 |
| | Step time | 0.22 | 0.65 |
| | Leading limb angle | 5.27 | 0.047 |
| | Trailing limb angle | 2.45 | 0.15 |
| | Peak propulsion | 0.06 | 0.81 |
| | Peak braking | 1.05 | 0.33 |
| | Propulsion impulse | 6.73 | 0.03 |
| | Braking impulse | 2.07 | 0.18 |
| **Factor: Time** | | | |
| | Step length | 0.23 | 0.65 |
| | Step time | 0.25 | 0.63 |
| | Leading limb angle | 0.98 | 0.35 |
| | Trailing limb angle | 0.55 | 0.48 |
| | Peak propulsion | 0.29 | 0.60 |
| | Peak braking | 0.02 | 0.90 |
| | Propulsion impulse | 1.02 | 0.34 |
| | Braking impulse | 0.07 | 0.80 |
| **Leg x Time Interaction** | | | |
| | Step length | 9.77 | 0.01 |
| | Step time | 3.33 | 0.10 |
| | Leading limb angle | 5.73 | 0.04 |
| | Trailing limb angle | 0.77 | 0.40 |
| | Peak propulsion | 2.99 | 0.12 |
| | Peak braking | 0.02 | 0.90 |
| | Propulsion impulse | 30.02 | <0.001 |
| | Braking impulse | 10.20 | 0.01 |
| | Post hoc Comparisons | | |
| | Step length | | |
| | *Baseline*, left vs. right | | 0.31 |
| | *Washout*, left vs. right | | 0.06 |
| | Left, *Baseline* vs. *Washout* | | 0.15 |
| | Right, *Baseline* vs. *Washout* | | 0.45 |
| | Leading limb angle | | |
| | *Baseline*, left vs. right | | 0.08 |
| | *Washout*, left vs. right | | 0.04 |
| | Left, *Baseline* vs. *Washout* | | 0.77 |
| | Right, *Baseline* vs. *Washout* | | 0.15 |
| | Propulsion impulse | | |
| | *Baseline*, left vs. right | | 0.49 |
| | *Washout*, left vs. right | | <0.01 |
| | Left, *Baseline* vs. *Washout* | | 0.01 |
| | Right, *Baseline* vs. *Washout* | | 0.06 |
| | Braking impulse | | |
| | *Baseline*, left vs. right | | 0.93 |
| | *Washout*, left vs. right | | 0.04 |
| | Left, *Baseline* vs. *Washout* | | 0.26 |
| | Right, *Baseline* vs. *Washout* | | 0.24 |

**Table 3. Statistical results for Experiment 2 open-loop *Fast* trial.**

| | Parameter | F(1,9) | p |
|---|---|---|---|
| **Factor: Leg** | | | |
| | Step length | 41.58 | <0.001 |
| | Step time | 0.81 | 0.39 |
| | Leading limb angle | 12.20 | <0.01 |
| | Trailing limb angle | 74.67 | <0.001 |
| | Peak propulsion | 52.86 | <0.001 |
| | Peak braking | 29.03 | <0.001 |
| | Propulsion impulse | 2.26 | 0.17 |
| | Braking impulse | 0.17 | 0.69 |
| **Factor: Time** | | | |
| | Step length | 0.96 | 0.35 |
| | Step time | 1.00 | 0.34 |
| | Leading limb angle | 4.67 | 0.06 |
| | Trailing limb angle | 5.91 | 0.04 |
| | Peak propulsion | 2.90 | 0.12 |
| | Peak braking | 0.55 | 0.48 |
| | Propulsion impulse | 0.08 | 0.79 |
| | Braking impulse | 0.03 | 0.86 |
| **Leg x Time Interaction** | | | |
| | Step length | 5.82 | 0.04 |
| | Step time | 1.57 | 0.24 |
| | Leading limb angle | 7.47 | 0.02 |
| | Trailing limb angle | 2.69 | 0.14 |
| | Peak propulsion | 0.01 | 0.93 |
| | Peak braking | 6.45 | 0.03 |
| | Propulsion impulse | 0.84 | 0.38 |
| | Braking impulse | 12.11 | 0.01 |
| | Post hoc Comparisons | | |
| | Step length | | |
| | Early epoch, left vs. right | | <0.01 |
| | Late epoch, left vs. right | | <0.001 |
| | Left, early vs. late epoch | | 0.045 |
| | Right, early vs. late epoch | | 0.04 |
| | Leading limb angle | | |
| | Early epoch, left vs. right | | <0.01 |
| | Late epoch, left vs. right | | 0.06 |
| | Left, early vs. late epoch | | <0.01 |
| | Right, early vs. late epoch | | 0.32 |
| | Peak braking | | |
| | Early epoch, left vs. right | | <0.01 |
| | Late epoch, left vs. right | | 0.02 |
| | Left, early vs. late epoch | | 0.047 |
| | Right, early vs. late epoch | | 0.06 |
| | Braking impulse | | |
| | Early epoch, left vs. right | | 0.07 |
| | Late epoch, left vs. right | | <0.01 |
| | Left, early vs. late epoch | | <0.01 |
| | Right, early vs. late epoch | | 0.02 |

**Table 4. Statistical results for Experiment 2 open-loop *Slow* trial.**

| | Parameter | F(1,9) | p |
|---|---|---|---|
| **Factor: Leg** | | | |
| | Step length | 83.14 | <0.001 |
| | Step time | 0.72 | 0.14 |
| | Leading limb angle | 5.62 | 0.04 |
| | Trailing limb angle | 299.71 | <0.001 |
| | Peak propulsion | 86.27 | <0.001 |
| | Peak braking | 5.94 | 0.04 |
| | Propulsion impulse | 2.79 | 0.13 |
| | Braking impulse | 15.81 | <0.01 |
| **Factor: Time** | | | |
| | Step length | 3.03 | 0.12 |
| | Step time | 0.30 | 0.60 |
| | Leading limb angle | 28.08 | <0.001 |
| | Trailing limb angle | 2.07 | 0.18 |
| | Peak propulsion | 1.00 | 0.35 |
| | Peak braking | 0.02 | 0.91 |
| | Propulsion impulse | 2.32 | 0.16 |
| | Braking impulse | 0.94 | 0.36 |
| **Leg x Time Interaction** | | | |
| | Step length | 2.21 | 0.17 |
| | Step time | 0.25 | 0.63 |
| | Leading limb angle | 4.04 | 0.08 |
| | Trailing limb angle | 0.21 | 0.66 |
| | Peak propulsion | 3.31 | 0.10 |
| | Peak braking | 7.74 | 0.02 |
| | Propulsion impulse | 0.00 | 0.99 |
| | Braking impulse | 18.96 | <0.01 |
| | Post hoc Comparisons | | |
| | Peak braking | | |
| | Early epoch, left vs. right | | <0.01 |
| | Late epoch, left vs. right | | 0.68 |
| | Left, early vs. late epoch | | 0.07 |
| | Right, early vs. late epoch | | 0.04 |
| | Braking impulse | | |
| | Early epoch, left vs. right | | 0.03 |
| | Late epoch, left vs. right | | <0.01 |
| | Left, early vs. late epoch | | 0.03 |
| | Right, early vs. late epoch | | <0.01 |

than right), and braking impulse (left larger than right) during the *Washout* trial that were not present during *Baseline*. The left leg also generated a significantly larger propulsion impulse during the *Washout* trial when compared to *Baseline*. Full statistical results for the Experiment 1 *Washout* vs. *Baseline* comparisons are shown in Table 2.

All summary statistics and statistical results for Experiment 1 are shown in Fig 4.

**Table 5. Statistical results for Experiment 2 open-loop *Accelerate* trial.**

|  | Parameter | F(1,9) | p |
|---|---|---|---|
| **Factor: Leg** |  |  |  |
|  | Step length | 0.04 | 0.85 |
|  | Step time | 64.63 | <0.001 |
|  | Leading limb angle | 47.66 | <0.001 |
|  | Trailing limb angle | 1.10 | 0.32 |
|  | Peak propulsion | 3.49 | 0.10 |
|  | Peak braking | 35.95 | <0.001 |
|  | Propulsion impulse | 11.05 | <0.01 |
|  | Braking impulse | 3.94 | 0.08 |
| **Factor: Time** |  |  |  |
|  | Step length | 2.47 | 0.15 |
|  | Step time | 2.84 | 0.13 |
|  | Leading limb angle | 0.27 | 0.62 |
|  | Trailing limb angle | 0.00 | 0.95 |
|  | Peak propulsion | 10.83 | <0.01 |
|  | Peak braking | 5.68 | 0.04 |
|  | Propulsion impulse | 2.17 | 0.18 |
|  | Braking impulse | 5.95 | 0.04 |
| **Leg x Time Interaction** |  |  |  |
|  | Step length | 0.22 | 0.65 |
|  | Step time | 0.95 | 0.36 |
|  | Leading limb angle | 4.52 | 0.06 |
|  | Trailing limb angle | 0.15 | 0.71 |
|  | Peak propulsion | 0.08 | 0.78 |
|  | Peak braking | 0.02 | 0.90 |
|  | Propulsion impulse | 0.55 | 0.48 |
|  | Braking impulse | 0.34 | 0.57 |

## Experiment 2

In Experiment 2, we tested whether we could drive selective asymmetric changes in gait using open-loop controllers that changed the treadmill speed mid-gait cycle but did not require input from force plates (as was required in Experiment 1). Participants' right heel-strikes were paced with a metronome to ensure that the treadmill speed changed at a specific time within the gait cycle. We tested two pairs of conditions: one pair where the treadmill speed changed over the middle half of the gait cycle (*Fast* and *Slow* trials) and another where the treadmill speed changed approximately at right heel-strike (*Accelerate* and *Decelerate* trials). Examples of a participant walking on the treadmill under each of the four conditions are shown in S1 Video. We also examined whether participants developed a preference for a particular walking pattern when the open-loop controller was activated but heel-strike timing was unconstrained after experiencing the spectrum of different possible walking patterns when guided by the metronome. As in Experiment 1, the participants walked relatively symmetrically at baseline with no significant differences in the asymmetry values for any gait parameters when compared to zero (all p > 0.05).

**Table 6. Statistical results for Experiment 2 open-loop *Decelerate* trial.**

|  | Parameter | F(1,9) | p |
|---|---|---|---|
| **Factor: Leg** |  |  |  |
|  | Step length | 7.01 | 0.03 |
|  | Step time | 31.76 | <0.001 |
|  | Leading limb angle | 0.53 | 0.49 |
|  | Trailing limb angle | 0.84 | 0.38 |
|  | Peak propulsion | 1.07 | 0.33 |
|  | Peak braking | 45.2 | <0.001 |
|  | Propulsion impulse | 24.07 | <0.01 |
|  | Braking impulse | 2.64 | 0.14 |
| **Factor: Time** |  |  |  |
|  | Step length | 1.65 | 0.23 |
|  | Step time | 1.85 | 0.21 |
|  | Leading limb angle | 2.74 | 0.13 |
|  | Trailing limb angle | 0.18 | 0.69 |
|  | Peak propulsion | 6.33 | 0.03 |
|  | Peak braking | 1.06 | 0.33 |
|  | Propulsion impulse | 8.41 | 0.02 |
|  | Braking impulse | 4.54 | 0.06 |
| **Leg x Time Interaction** |  |  |  |
|  | Step length | 1.26 | 0.29 |
|  | Step time | 0.14 | 0.72 |
|  | Leading limb angle | 5.02 | 0.052 |
|  | Trailing limb angle | 0.46 | 0.51 |
|  | Peak propulsion | 0.16 | 0.70 |
|  | Peak braking | 5.12 | 0.050 |
|  | Propulsion impulse | 2.10 | 0.18 |
|  | Braking impulse | 1.93 | 0.20 |

## *Fast* trial

When participants walked with the open-loop controller activated during the *Fast* trial (Fig 5A), we observed that they walked with significant asymmetries in step length (right larger than left; Fig 5B), leading limb angles (right larger than left; Fig 5C), and trailing limb angles (left larger than right; Fig 5C). Step lengths and leading limb angles became significantly more symmetric–but remained significantly asymmetric–over the course of the *Fast* trial. Step times were not significantly asymmetric. Participants showed significant asymmetries in peak propulsive (left larger than right; 5D) and peak braking (right larger than left) forces, and peak braking forces became more symmetric while braking impulses became asymmetric over the course of the *Fast* trial. Propulsion impulses were not significantly asymmetric. Full statistical results for the Experiment 2 *Fast* trial are shown in Table 3.

## *Slow* trial

The *Slow* trial was designed to be the inverse of the *Fast* trial (Fig 6A) to test the robustness of our findings in the *Fast* trial (i.e., we expected the results to be opposite those observed in the *Fast* trial). We observed that participants walked with significant asymmetries in step length (left larger than right; Fig 6B), leading limb angles (right larger than left; Fig 6C), and trailing limb angles (right larger than left; Fig 6C). Step times were not significantly asymmetric.

**Table 7. Statistical results for Experiment 2 asymmetry comparisons across *Metronome* trials.**

| | Parameter | F(3,27) | p |
|---|---|---|---|
| **Factor: Trial** | | | |
| | Step length asymmetry | 19.97 | <0.001 |
| | Step time asymmetry | 19.88 | <0.001 |
| | Leading limb angle asymmetry | 15.72 | <0.001 |
| | Trailing limb angle asymmetry | 34.76 | <0.001 |
| | Peak propulsion asymmetry | 32.57 | <0.001 |
| | Peak braking asymmetry | 30.64 | <0.001 |
| | Propulsion impulse asymmetry | 17.4 | <0.001 |
| | Braking impulse asymmetry | 0.95 | 0.43 |
| **Factor: Time** | | | |
| | Step length asymmetry | 4.92 | 0.05 |
| | Step time asymmetry | 0.01 | 0.94 |
| | Leading limb angle asymmetry | 7.32 | 0.02 |
| | Trailing limb angle asymmetry | 2.28 | 0.17 |
| | Peak propulsion asymmetry | 0.01 | 0.93 |
| | Peak braking asymmetry | 0.89 | 0.37 |
| | Propulsion impulse asymmetry | 0.02 | 0.90 |
| | Braking impulse asymmetry | 0.78 | 0.40 |
| **Trial x Time Interaction** | | | |
| | Step length asymmetry | 2.33 | 0.10 |
| | Step time asymmetry | 0.96 | 0.43 |
| | Leading limb angle asymmetry | 5.55 | <0.01 |
| | Trailing limb angle asymmetry | 0.63 | 0.61 |
| | Peak propulsion asymmetry | 1.02 | 0.40 |
| | Peak braking asymmetry | 2.08 | 0.13 |
| | Propulsion impulse asymmetry | 1.11 | 0.36 |
| | Braking impulse asymmetry | 5.11 | <0.01 |
| | Post hoc Comparisons (main effect of Trial) | | |
| | Step length asymmetry | | |
| | *Accelerate* vs. *Fast* | | <0.001 |
| | *Accelerate* vs. *Decelerate* | | 0.07 |
| | *Accelerate* vs. *Slow* | | 0.04 |
| | *Fast* vs. *Decelerate* | | 0.02 |
| | *Fast* vs. *Slow* | | <0.001 |
| | *Decelerate* vs. *Slow* | | <0.001 |
| | Step time asymmetry | | |
| | *Accelerate* vs. *Fast* | | <0.01 |
| | *Accelerate* vs. *Decelerate* | | <0.001 |
| | *Accelerate* vs. *Slow* | | <0.001 |
| | *Fast* vs. *Decelerate* | | <0.01 |
| | *Fast* vs. *Slow* | | 0.34 |
| | *Decelerate* vs. *Slow* | | 0.01 |
| | Trailing limb angle asymmetry | | |
| | *Accelerate* vs. *Fast* | | <0.001 |
| | *Accelerate* vs. *Decelerate* | | 0.78 |
| | *Accelerate* vs. *Slow* | | <0.001 |
| | *Fast* vs. *Decelerate* | | <0.001 |

*(Continued)*

**Table 7.** (Continued)

| | Parameter | F(3,27) | p |
|---|---|---|---|
| | *Fast* vs. *Slow* | | <0.001 |
| | *Decelerate* vs. *Slow* | | <0.001 |
| | Peak propulsion asymmetry | | |
| | *Accelerate* vs. *Fast* | | <0.001 |
| | *Accelerate* vs. *Decelerate* | | 0.02 |
| | *Accelerate* vs. *Slow* | | 0.02 |
| | *Fast* vs. *Decelerate* | | <0.01 |
| | *Fast* vs. *Slow* | | <0.001 |
| | *Decelerate* vs. *Slow* | | <0.001 |
| | Peak braking asymmetry | | |
| | *Accelerate* vs. *Fast* | | 0.99 |
| | *Accelerate* vs. *Decelerate* | | <0.001 |
| | *Accelerate* vs. *Slow* | | <0.001 |
| | *Fast* vs. *Decelerate* | | <0.001 |
| | *Fast* vs. *Slow* | | <0.001 |
| | *Decelerate* vs. *Slow* | | 0.99 |
| | Propulsion impulse asymmetry | | |
| | *Accelerate* vs. *Fast* | | <0.001 |
| | *Accelerate* vs. *Decelerate* | | <0.001 |
| | *Accelerate* vs. *Slow* | | 0.08 |
| | *Fast* vs. *Decelerate* | | 0.08 |
| | *Fast* vs. *Slow* | | 0.04 |
| | *Decelerate* vs. *Slow* | | <0.001 |
| | Post hoc Comparisons (Trial x Time interaction) | | |
| | Leading limb angle asymmetry | | |
| | *Accelerate*, early vs. *Fast*, early | | 1.00 |
| | *Accelerate*, early vs. *Decelerate*, early | | <0.001 |
| | *Accelerate*, early vs. *Slow*, early | | <0.001 |
| | *Accelerate*, early vs. *Accelerate*, late | | 0.86 |
| | *Accelerate*, early vs. *Fast*, late | | <0.01 |
| | *Accelerate*, early vs. *Decelerate*, late | | <0.001 |
| | *Accelerate*, early vs. *Slow*, late | | 0.01 |
| | *Fast*, early vs. *Decelerate*, early | | <0.01 |
| | *Fast*, early vs. *Slow*, early | | <0.01 |
| | *Fast*, early vs. *Accelerate*, late | | 1.00 |
| | *Fast*, early vs. *Fast*, late | | <0.01 |
| | *Fast*, early vs. *Decelerate*, late | | <0.001 |
| | *Fast*, early vs. *Slow*, late | | 0.13 |
| | *Decelerate*, early vs. *Slow*, early | | 1.00 |
| | *Decelerate*, early vs. *Accelerate*, late | | <0.01 |
| | *Decelerate*, early vs. *Fast*, late | | 1.00 |
| | *Decelerate*, early vs. *Decelerate*, late | | 0.44 |
| | *Decelerate*, early vs. *Slow*, late | | 1.00 |
| | *Slow*, early vs. *Accelerate*, late | | 0.02 |
| | *Slow*, early vs. *Fast*, late | | 1.00 |
| | *Slow*, early vs. *Decelerate*, late | | 0.45 |
| | *Slow*, early vs. *Slow*, late | | 1.00 |

(*Continued*)

**Table 7.** (Continued)

| | Parameter | F(3,27) | p |
|---|---|---|---|
| | *Accelerate*, late vs. *Fast*, late | | 0.10 |
| | *Accelerate*, late vs. *Decelerate*, late | | <0.001 |
| | *Accelerate*, late vs. *Slow*, late | | 0.20 |
| | *Fast*, late vs. *Decelerate*, late | | 0.10 |
| | *Fast*, late vs. *Slow*, late | | 1.00 |
| | *Decelerate*, late vs. *Slow*, late | | 0.04 |
| Braking impulse asymmetry | | | |
| | No statistically significant post hoc comparisons. | | |

Participants showed significant asymmetries in peak propulsion (right larger than left; Fig 6D) and braking impulses (right larger than left; Fig 6D). Braking impulses become more asymmetric throughout the *Slow* trial. Propulsion impulses were not significantly asymmetric. Full statistical results for the Experiment 2 *Slow* trial are shown in Table 4.

> **Accelerate trial.** When participants walked with the open-loop controller activated during the *Accelerate* trial (Fig 7A), we observed that they walked with significant asymmetries in step

**Table 8. Statistical results for Experiment 2 *Pre* vs. *Post* comparisons.**

| | Parameter | F(1,9) | p |
|---|---|---|---|
| **Factor: Leg** | | | |
| | Step length | 0.01 | 0.94 |
| | Step time | 0.04 | 0.85 |
| | Leading limb angle | 2.22 | 0.17 |
| | Trailing limb angle | 0.16 | 0.70 |
| | Peak propulsion | 1.46 | 0.26 |
| | Peak braking | 0.18 | 0.68 |
| | Propulsion impulse | 0.71 | 0.42 |
| | Braking impulse | 4.14 | 0.07 |
| **Factor: Time** | | | |
| | Step length | 4.57 | 0.06 |
| | Step time | 3.39 | 0.10 |
| | Leading limb angle | 5.35 | 0.046 |
| | Trailing limb angle | 2.75 | 0.13 |
| | Peak propulsion | 13.45 | <0.01 |
| | Peak braking | 0.40 | 0.54 |
| | Propulsion impulse | 6.46 | 0.03 |
| | Braking impulse | 2.77 | 0.13 |
| **Leg x Time Interaction** | | | |
| | Step length | 4.40 | 0.07 |
| | Step time | 0.00 | 1.00 |
| | Leading limb angle | 0.63 | 0.45 |
| | Trailing limb angle | 0.86 | 0.38 |
| | Peak propulsion | 1.22 | 0.30 |
| | Peak braking | 0.40 | 0.54 |
| | Propulsion impulse | 0.89 | 0.37 |
| | Braking impulse | 4.05 | 0.08 |

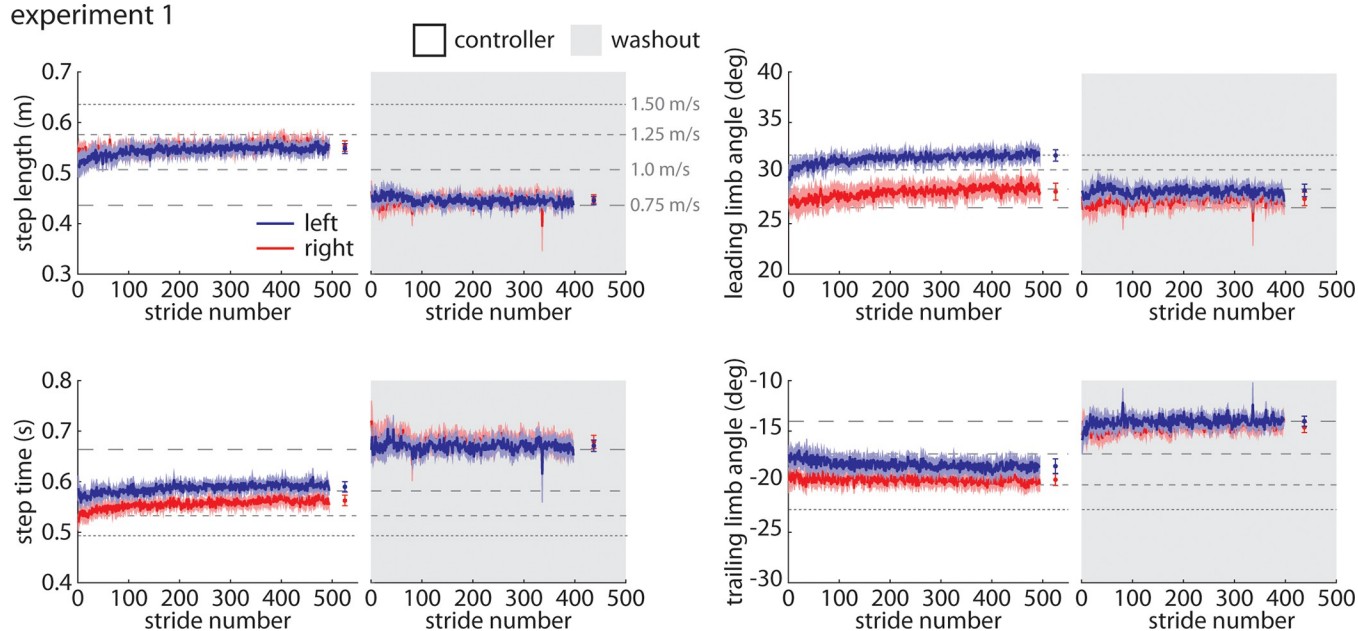

**Fig 2. Spatiotemporal and kinematic parameters during the controller (white) and washout (gray) trials during Experiment 1.** The blue line indicates the group mean±SEM for the left leg and the red line indicates the group mean ±SEM for the right leg. Statistical results are included in Tables 1 and 2 and shown in Fig 4 to preserve clarity of the time-series plots.

time (left larger than right; Fig 7B) and leading limb angles (right larger than left; Fig 7C). Step lengths and trailing limb angles were not significantly asymmetric. Participants showed significant asymmetries in peak braking and propulsion impulse (right larger than left; Fig 7D). Full statistical results for the Experiment 2 *Accelerate* trial are shown in Table 5.

*Decelerate* **trial.** The *Decelerate* trial was designed to be the inverse of the *Accelerate* trial–participants walked with the open-loop controller activated and the treadmill moved slow over the first ~50% of the gait cycle and fast over the last ~50% of the gait cycle (Fig 8A). We observed that participants walked with significant asymmetries in step length and step time (right larger than left; Fig 8B). Leading and trailing limb angles were not significantly asymmetric (Fig 8C. Participants showed significant asymmetries in peak braking and propulsion impulse (left larger than right; Fig 8D). Full statistical results for the Experiment 2 *Decelerate* trial are shown in Table 6.

Time series AP GRF profiles (group mean ± SEM) for the first and last five strides of all four trials (*Fast*, *Slow*, *Accelerate*, and *Decelerate)* are shown in Fig 9 for direct comparison among trials and over time.

**Gait asymmetries across** *Metronome trials.* Next, we compared the asymmetries in each gait parameter directly across the four *Metronome* trials and over time. Due to the large number of comparisons, the full statistical results are shown in Table 7. In short, these results largely confirm the findings that one would expect when comparing across the *Metronome* trials described above. We also show asymmetry data for all gait parameters in Fig 10.

*Pre* **and** *Post* **trials.** Finally, we compared the *Pre* and *Post* trials to assess whether participants developed preferences for particular walking patterns when walking with the open-loop controller engaged (and no metronome) after experiencing the metronome-paced *Fast*, *Slow*, *Accelerate*, and *Decelerate* trials. In both the *Pre* (Fig 11A) and *Post* (Fig 11B) trials, step lengths, step times, leading limb angles, trailing limb angles, peak propulsion forces, peak

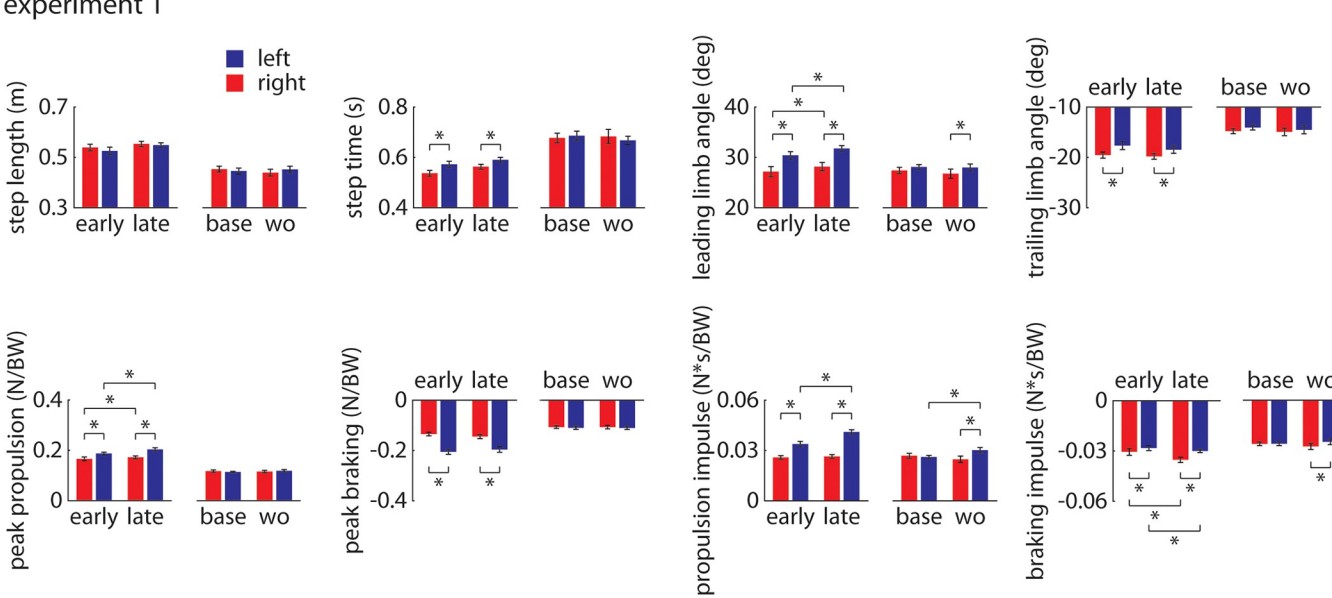

**Fig 3. Kinetic parameters during the controller (white) and washout (gray) trials during Experiment 1.** Color scheme follows that of Fig 2. Statistical results are included in Tables 1 and 2 and shown in Fig 4 to preserve clarity of the time-series plots.

**Fig 4. Results of statistical tests for Experiment 1 comparing gait parameters (group mean±SEM) between legs and between time points.** 'Base' indicates data from the second 0.75 m/s *Baseline* trial and 'wo' indicates data from the *Washout* trial. * indicates p<0.05.

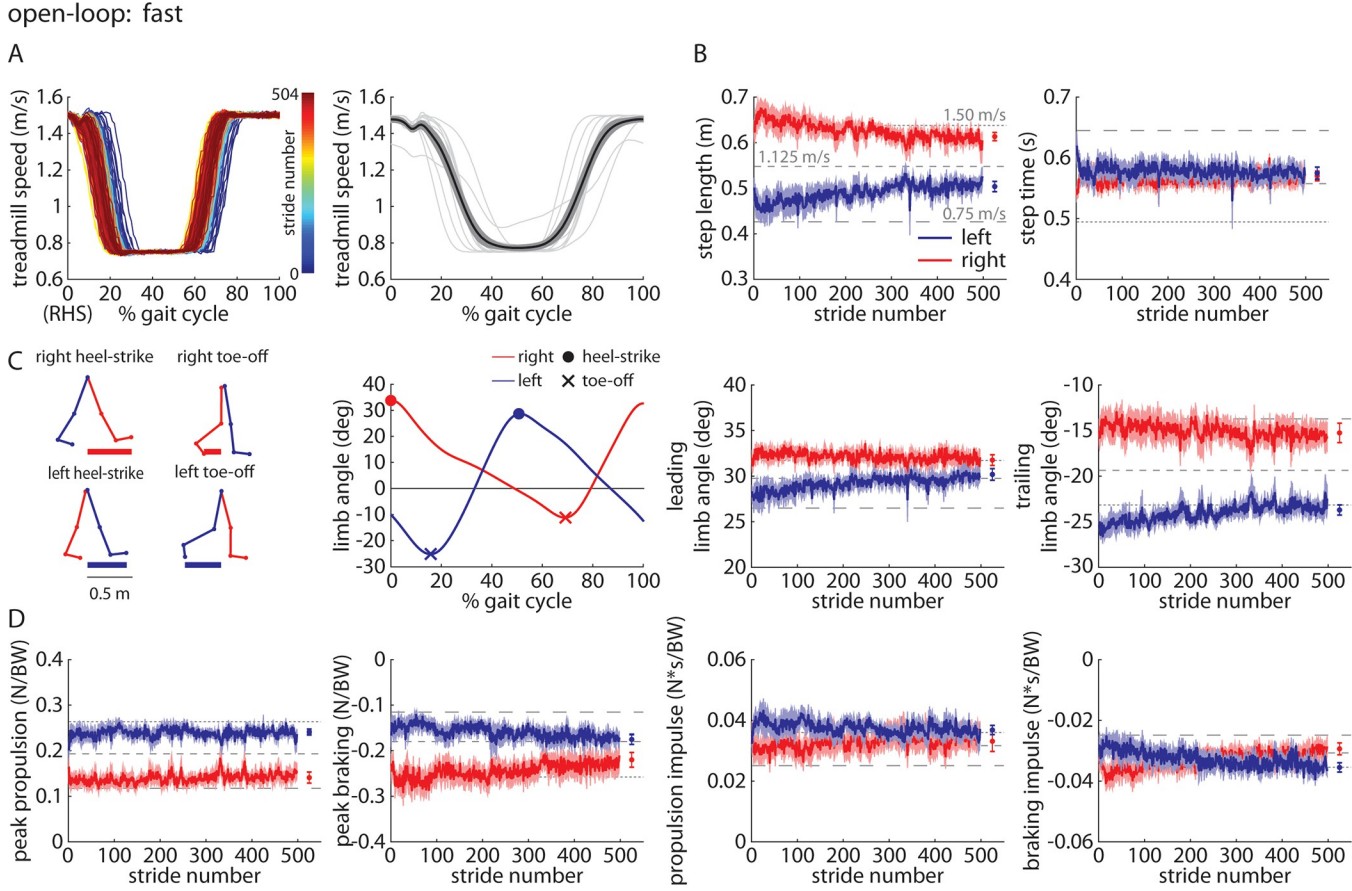

**Fig 5.** Time-series plots of A) treadmill speed (representative participant shown on left, group mean (black) ±SEM and individual participant means (gray) shown on the right, B) group mean ±SEM for the left (blue) and right (red) step lengths and step times, where horizontal dashed lines indicate the group mean of each of the three *Baseline* trials for reference, C) kinematic parameters for each leg, and D) kinetic parameters for each leg during the *Fast* trial of Experiment 2. Statistical results are included in Table 3 and Fig 12.

braking forces, propulsion impulses, and braking impulses were not significantly asymmetric. Leading limb angles and peak propulsive forces were significantly larger bilaterally in the *Post* trial when compared to the *Pre* trial. Full statistical results for the Experiment 2 *Pre* vs. *Post* comparisons are shown in Table 8.

All summary statistics and statistical results for Experiment 2 are shown in Fig 12.

## Discussion

In this study, we demonstrated that simple within-stride changes in treadmill speed can drive selective changes in human gait symmetry. We found that, in support of our hypotheses, we could drive asymmetric changes in a variety of different kinematic, kinetic, and spatiotemporal gait parameters by accelerating or decelerating the treadmill speed at different points within the gait cycle. We observed that this could be achieved using either closed-loop or open-loop treadmill control, with the open-loop approach requiring an external device (i.e., a metronome) to pace the participant's footfalls to achieve changes in gait symmetry.

There is a clear need for customizable rehabilitation approaches that can be tailored to the needs of each individual patient and delivered using accessible equipment. Previous approaches have aimed to restore gait symmetry using specialized treadmills [22–24], robotics

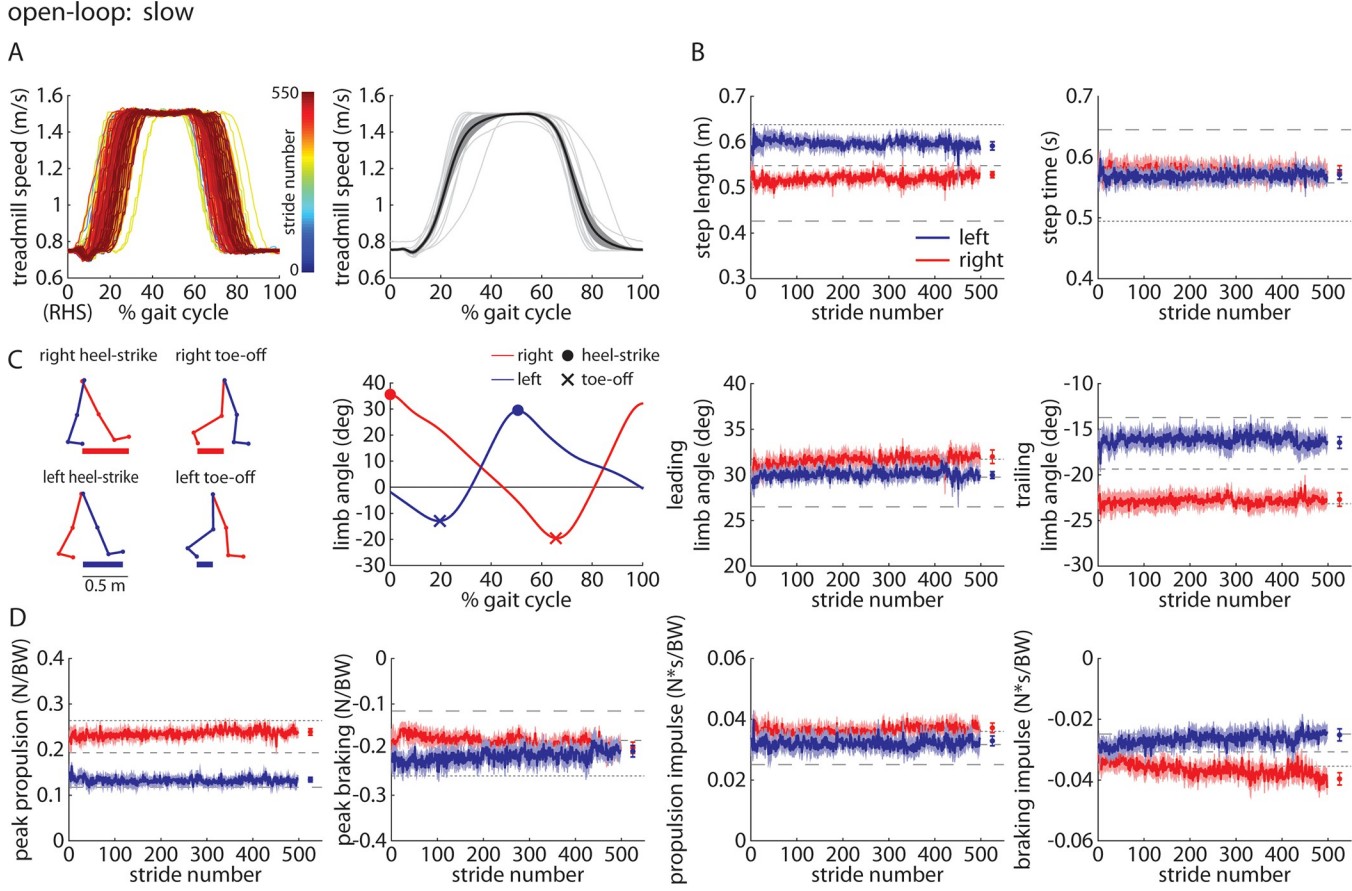

**Fig 6.** Time-series plots of A) treadmill speed (representative participant shown on left, group mean (black) ±SEM and individual participant means (gray) shown on the right, B) group mean±SEM for the left (blue) and right (red) step lengths and step times, where horizontal dashed lines indicate the group mean of each of the three *Baseline* trials for reference, C) kinematic parameters for each leg, and D) kinetic parameters for each leg during the *Slow* trial of Experiment 2. Statistical results are included in Table 4 and Fig 12.

[25–29], electrical stimulation [30–32], other technologies (e.g., biofeedback; [33–35]), and conventional gait training [36]. Here, we observed that we could drive changes in the asymmetry of specific gait parameters (in healthy adults) using a tied-belt treadmill with no need for external feedback or instruction from an experimenter or clinician. Accordingly, this study was conducted as a first step toward the development of a simple treadmill-based approach that could be customized to address a variety of gait asymmetries in different clinical populations (e.g., stroke, cerebral palsy, lower limb amputation).

It is important to consider the mechanism by which a rehabilitation approach drives changes in gait. For example, the mismatch between the belt speeds during split-belt treadmill walking induces a mechanism of motor learning termed 'adaptation' that results in immediate short-term aftereffects (i.e., the newly learned gait pattern persists into one's everyday gait pattern immediately following a bout of training [22, 50]) that may become more permanent over time [37, 51]. Other approaches (e.g., biofeedback, fast treadmill walking) drive immediate changes in gait during training but do not result in robust short-term aftereffects. Instead, these types of approaches are designed to elicit cumulative, long-term changes in everyday gait through repetitive training. The approach used in this study appears to align with the latter; while we did observe immediate aftereffects in some parameters, these were relatively small in magnitude. We anticipate that the potential rehabilitation value from this approach would

open-loop: accelerate

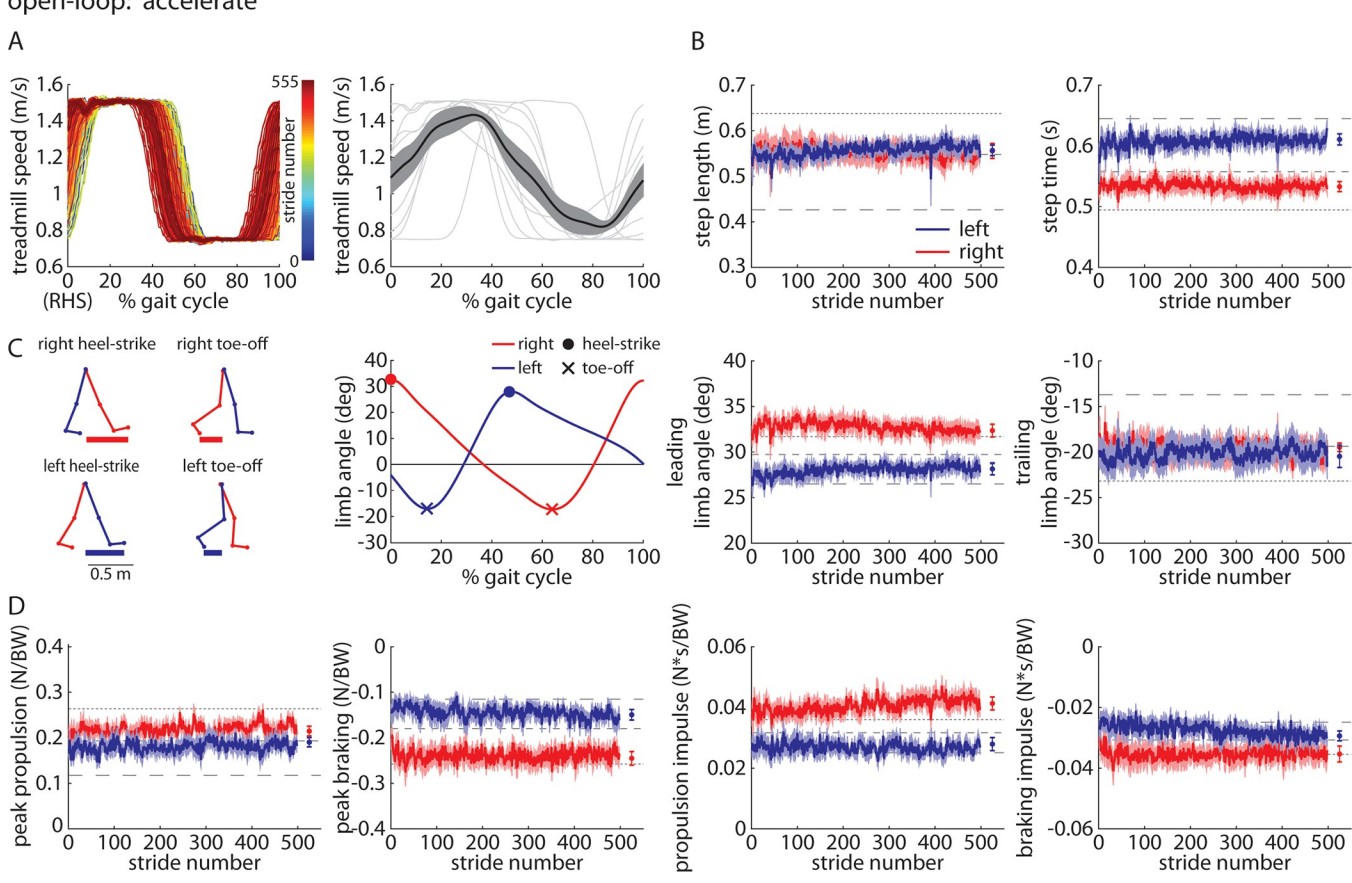

**Fig 7.** Time-series plots of A) treadmill speed (representative participant shown on left, group mean (black) ±SEM and individual participant means (gray) shown on the right, B) group mean±SEM for the left (blue) and right (red) step lengths and step times, where horizontal dashed lines indicate the group mean of each of the three *Baseline* trials for reference, C) kinematic parameters for each leg, and D) kinetic parameters for each leg during the *Accelerate* trial of Experiment 2. Statistical results are included in Table 5 and Fig 12.

result from cumulative effects of repetitive training rather than short-term adaptation with immediate aftereffects.

While some of the findings observed in Experiments 1 and 2 are likely related directly to the treadmill speed (e.g., the foot was placed farther ahead of the body when stepping onto a fast moving treadmill regardless of the timing of the treadmill speed change), others were more unexpected. For example, the propulsion impulses observed in the left leg of the Experiment 1 *Controller* trial, right leg of the Experiment 2 *Accelerate* trial, and left leg of the Experiment 2 *Decelerate* trial were markedly larger than those observed during even the fastest baseline walking trial. In other words, the within-stride changes in treadmill speed could drive larger propulsive impulses (unilaterally) than fast treadmill walking, a common intervention for persons post-stroke [52]. This may have particular relevance for rehabilitation given the strong interest in restoring propulsion unilaterally in patients with gait asymmetry [39, 40]. While future research into potential effects of long-term gait training with this approach are needed, we could speculate, for example, a customized protocol for an individual with stroke with unilateral hemiparesis resulting in reduced forward propulsion in one leg. In this customizable protocol, we might advise the open-loop (Experiment 2) controller *Slow*, which differentially increased trailing limb angle and increased propulsion (Fig 5). Still, more research is

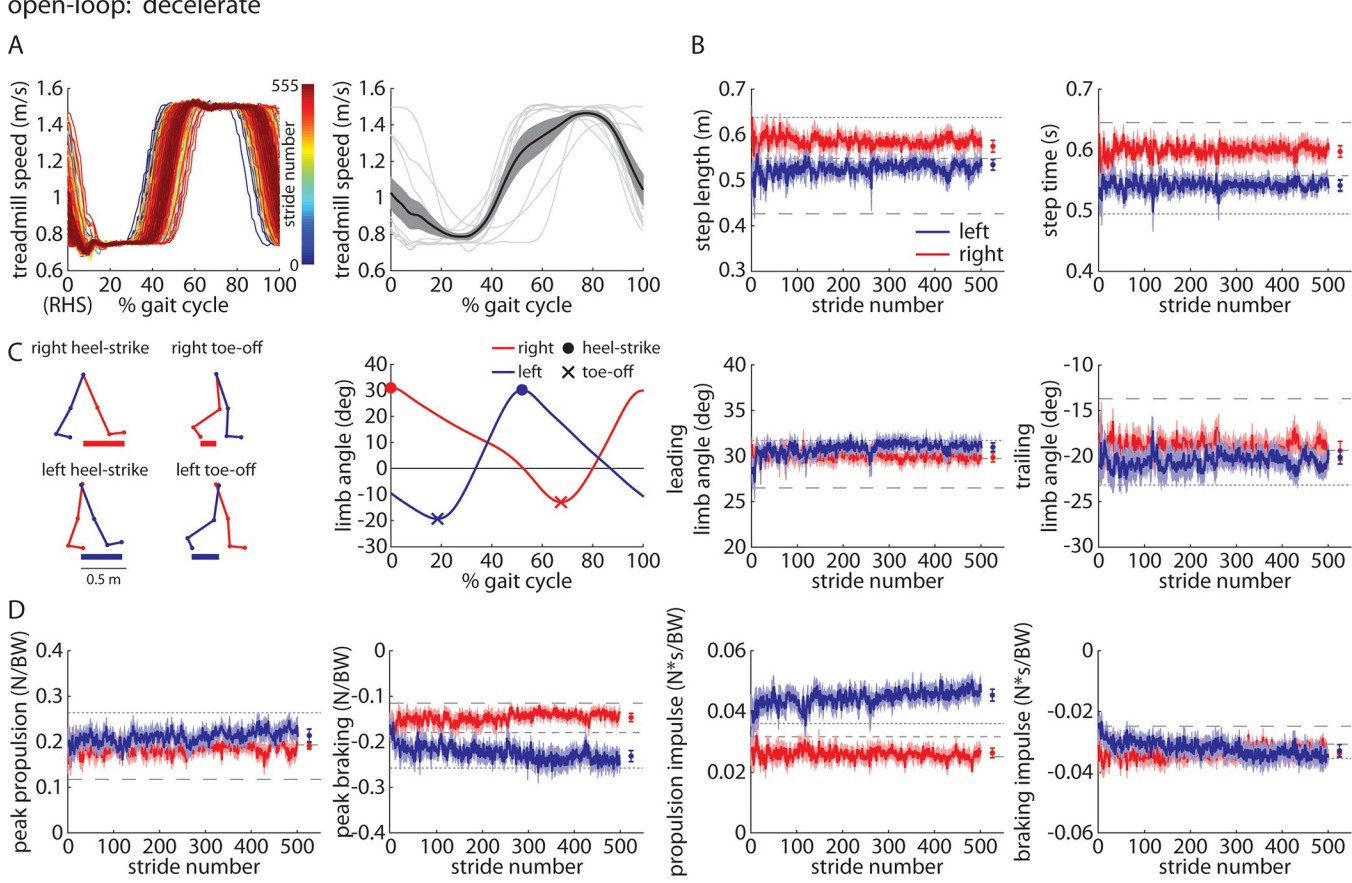

**Fig 8.** Time-series plots of A) treadmill speed (representative participant shown on left, group mean (black) ±SEM and individual participant means (gray) shown on the right, B) group mean±SEM for the left (blue) and right (red) step lengths and step times, where horizontal dashed lines indicate the group mean of each of the three *Baseline* trials for reference, C) kinematic parameters for each leg, and D) kinetic parameters for each leg during the *Decelerate* trial of Experiment 2. Statistical results are included in Table 6 and Fig 12.

needed to understand the implications of this approach for gait training in clinical populations.

In Experiment 2, we also aimed to understand whether participants would develop preferences for particular gait patterns when walking with the open-loop controller engaged but no metronome to pace their footfalls. Ultimately, participants did not show a consistent preference for any specific walking pattern during the *Pre* trial or the *Post* trial, despite the 40 minutes of experience walking with the different metronome-paced conditions. We anecdotally observed participants responding in a few different ways that were ultimately unable to be assessed statistically with this design. It appeared that some participants volitionally synchronized with their "preferred" modality for short durations (e.g., *Fast*) but switched to other synchronized heel-strikes either deliberately or as a result of natural variability in stride time. Still, others never seemed to synchronize for more than a few steps at a time without the metronome pacing to guide heel-strikes. It is possible that such a preference may develop with additional experience walking with the open-loop controller engaged over longer timescales, but this was not explored in this single session study.

Though this study demonstrates that within-stride changes in treadmill speed can drive changes in gait asymmetry, this approach necessitates 1) a treadmill controller capable of modulating the treadmill speed within precise time intervals and 2) a treadmill motor with

experiment 2

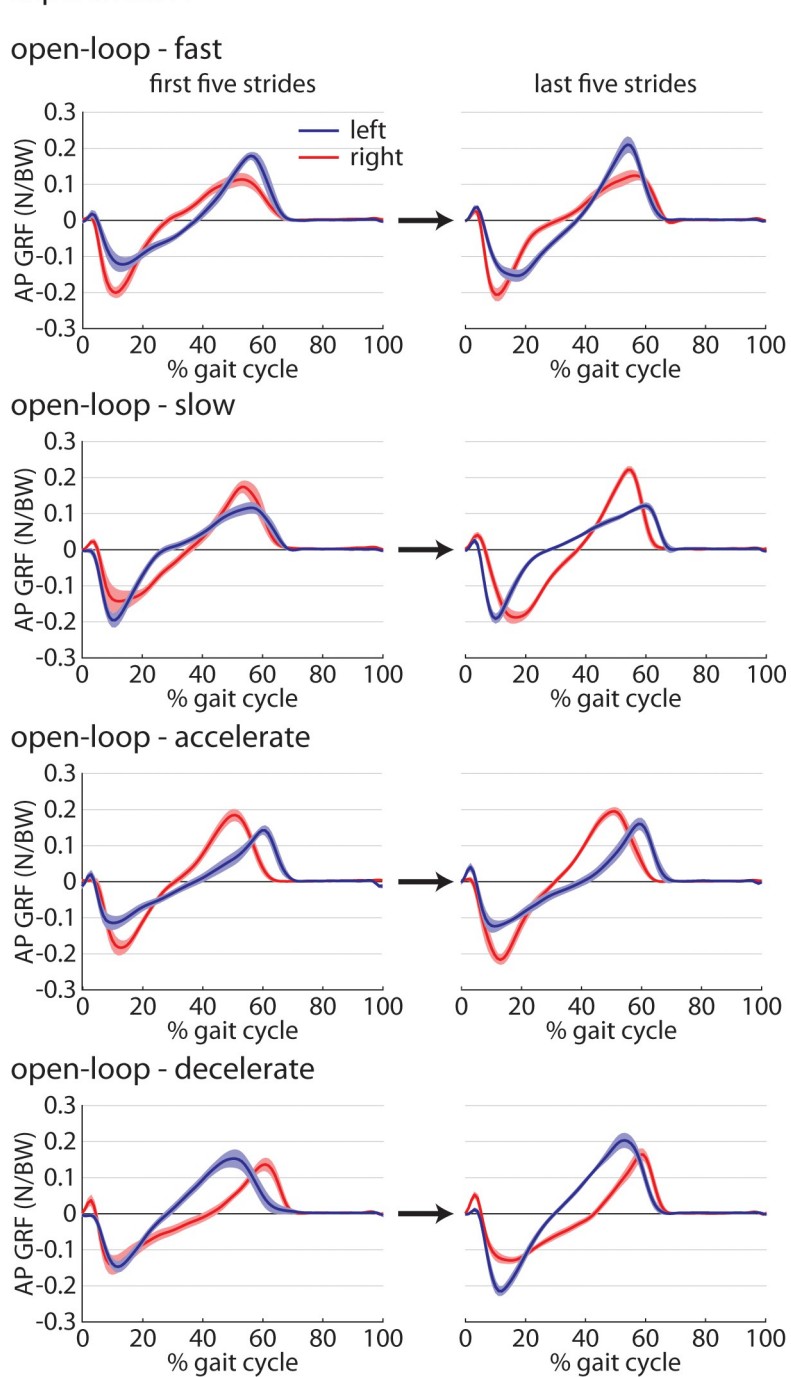

**Fig 9.** Group mean±SEM time-series plots of anteroposterior ground reaction force (AP GRF) data across the first (left) and last (right) five gait cycles for each of the four *Metronome* trials (*Fast, Slow, Accelerate, and Decelerate*).

sufficient acceleration and deceleration capabilities to drive these within-stride changes. The specifications are not available on most treadmills found in homes, gyms, or rehabilitation centers. Therefore, new treadmills beyond those currently available in most standard home or

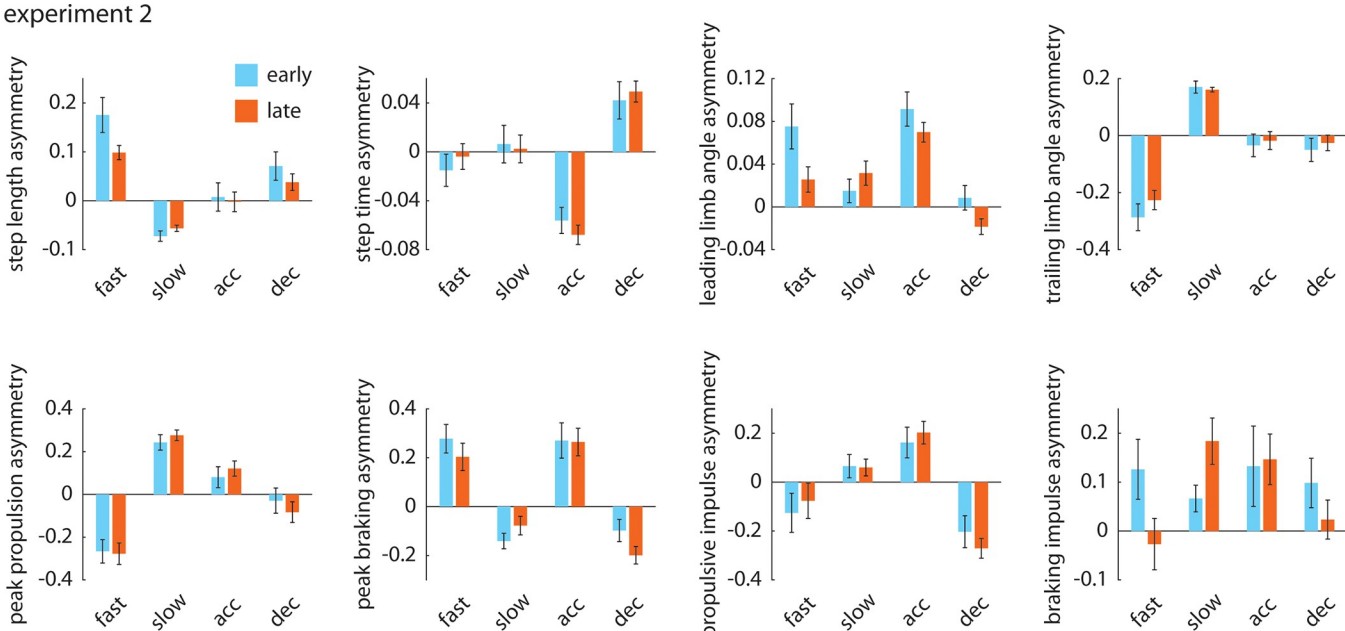

**Fig 10. Group mean±SEM asymmetric metrics for each of the gait parameters measured across the four *Metronome* trials (*Fast, Slow, Accelerate, and Decelerate*) in Experiment 2.** We show both early (cyan) and late (orange) values. To preserve figure clarity, indicators of statistical significance are omitted; we provide full results of the statistical analyses in Table 7.

clinical settings will need to be designed to enable this approach. Indeed, active research efforts are underway to develop such a treadmill to facilitate improved accessibility to this approach.

There are notable limitations to this study. While we think that this approach may eventually have clinical utility, only healthy adults were tested in these experiments. It is not clear that patients with various patterns of gait asymmetry and/or different levels of motor function would change their walking similarly to healthy adults when exposed to this treadmill approach. Due to the ongoing COVID-19 pandemic, we did not collect metabolic data or other measures of energy expenditure in this experiment. These data would be helpful in providing insight into the energy demand associated with the various different walking conditions. We also did not test gait parameters beyond the mean values; it is possible that this new treadmill walking approach also affects other clinically relevant aspects of gait (e.g., gait variability). Finally, we only tested four conditions using the open-loop treadmill controller. It is possible that other conditions where the treadmill speed changes at untested points within the gait cycle or with variable accelerations may provide additional options for targeting asymmetry in specific gait parameters.

## Conclusions

Here, we observed that simple within-stride changes in treadmill speed can drive selective changes in human gait symmetry. The timing of the speed change within the gait cycle dictated which gait parameters became asymmetric or remained symmetric. We were able to drive changes in the symmetry of a variety of different, clinically relevant gait parameters including step length, step time, leading and trailing limb angles, and propulsive forces. Future work will aim to use this customizable, targeted approach to restore symmetry in specific gait parameters in patients with asymmetric walking patterns.

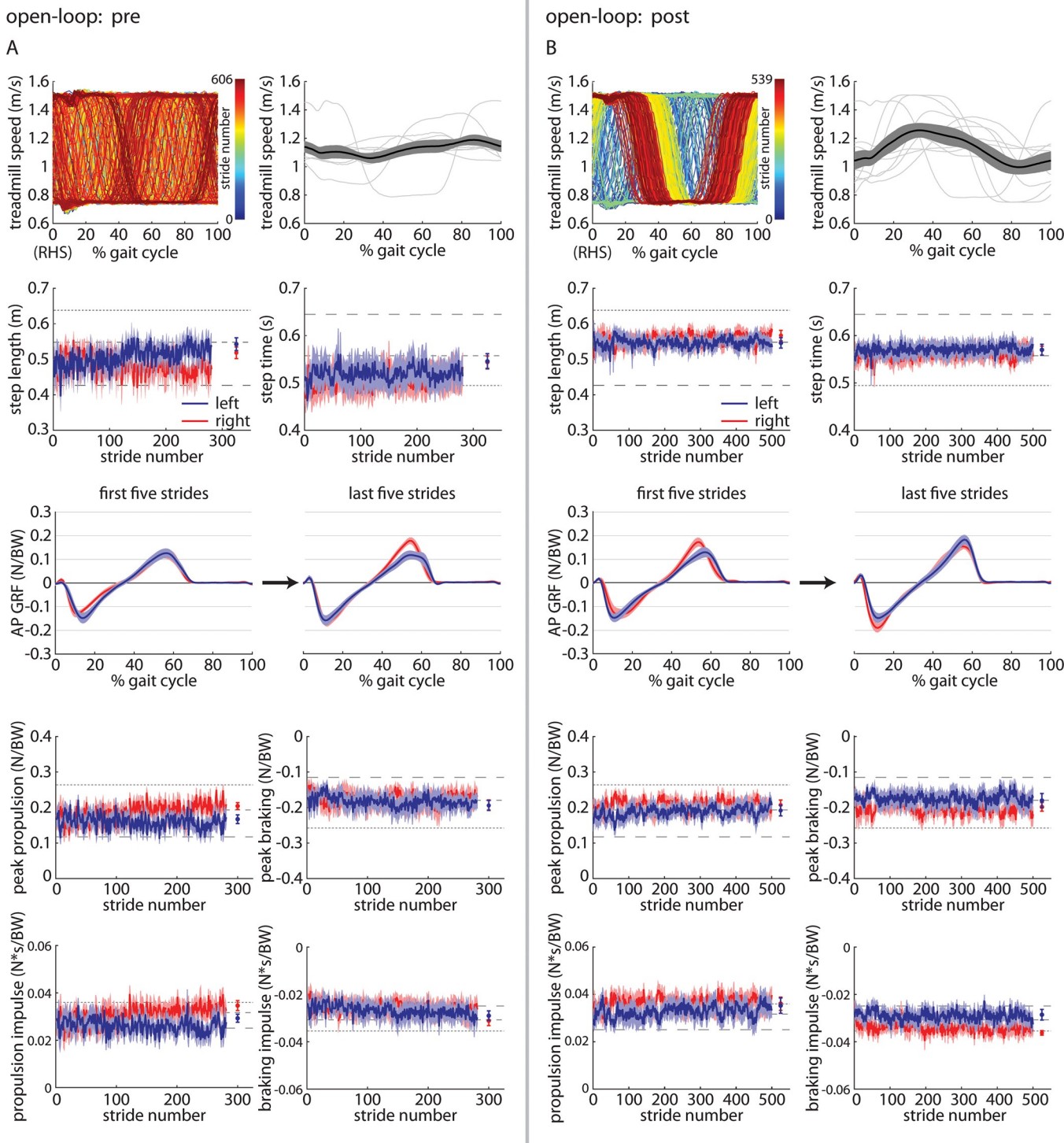

**Fig 11.** Time-series plots of treadmill speed (representative participant shown on left, group mean (black) ±SEM and individual participant means (gray) shown on the right, group mean±SEM for the left (blue) and right (red) step lengths and step times, where horizontal dashed lines indicate the group mean of each of the three *Baseline* trials for reference, kinematic parameters for each leg, and kinetic parameters for each leg during the A) *Pre* and B) *Post* trials of Experiment 2.

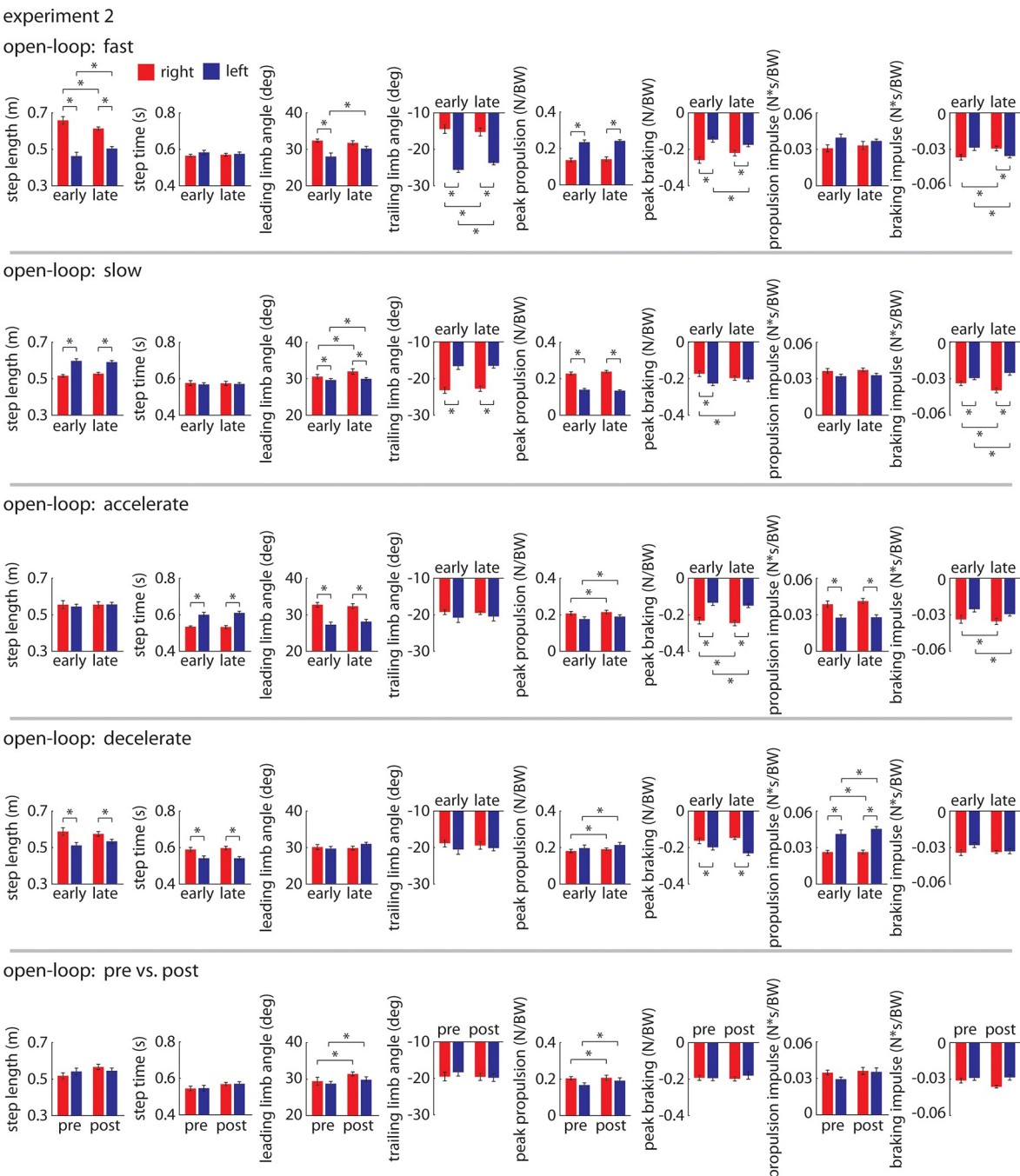

**Fig 12. Results of statistical tests for Experiment 2 comparing gait parameters (group mean±SEM) between legs and between early and late epochs across the four *Metronome* trials (*Fast, Slow, Accelerate, Decelerate*) and the two open-loop trials without metronome pacing (*Pre* and *Post*).** * indicates p<0.05.

## Supporting information

**S1 Video. Demonstration of a healthy young adult walking under the conditions tested in Experiment 2.** Author name: Ryan T. Roemmich, PhD. Videographer: Jan Stenum, PhD. Participant: Ryan T. Roemmich, PhD. Length: 1:51. Size: 77,700 MB.
(MP4)

## Author Contributions

**Conceptualization:** Michael G. Browne, Purnima Padmanabhan, Ryan T. Roemmich.

**Data curation:** Michael G. Browne, Jan Stenum.

**Formal analysis:** Michael G. Browne, Jan Stenum.

**Funding acquisition:** Michael G. Browne, Ryan T. Roemmich.

**Investigation:** Michael G. Browne, Jan Stenum, Purnima Padmanabhan, Ryan T. Roemmich.

**Methodology:** Michael G. Browne, Ryan T. Roemmich.

**Project administration:** Ryan T. Roemmich.

**Resources:** Ryan T. Roemmich.

**Software:** Michael G. Browne, Ryan T. Roemmich.

**Supervision:** Ryan T. Roemmich.

**Visualization:** Michael G. Browne, Ryan T. Roemmich.

**Writing – original draft:** Michael G. Browne, Ryan T. Roemmich.

**Writing – review & editing:** Michael G. Browne, Jan Stenum, Purnima Padmanabhan, Ryan T. Roemmich.

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
