## [Decision Letter · Decision Letter 0]

28 Oct 2022

PONE-D-22-20229Simple within-stride changes in treadmill speed can drive selective changes in human gait symmetryPLOS ONE

Dear Dr. Roemmich,

Thank you for submitting your manuscript to PLOS ONE. After careful consideration, we feel that it has merit but does not fully meet PLOS ONE’s publication criteria as it currently stands. Therefore, we invite you to submit a revised version of the manuscript that addresses the points raised during the review process.

We look forward to receiving your revised manuscript.

Kind regards,

Pei-Chun Kao

Academic Editor

PLOS ONE

Journal Requirements:

This project was funded by an American Heart Association Postdoctoral Fellowship (20POST35110071) to MGB, NIH grant R21 AG059184 to RTR, and an American Heart Association Career Development Award to RTR (935556).

Reviewers' comments:

Reviewer's Responses to Questions

**Comments to the Author**

1. Is the manuscript technically sound, and do the data support the conclusions?

Reviewer #1: Yes

Reviewer #2: Partly

2. Has the statistical analysis been performed appropriately and rigorously? 

Reviewer #1: No

Reviewer #2: Yes

3. Have the authors made all data underlying the findings in their manuscript fully available?

Reviewer #1: Yes

Reviewer #2: Yes

4. Is the manuscript presented in an intelligible fashion and written in standard English?

Reviewer #1: Yes

Reviewer #2: Yes

5. Review Comments to the Author

Reviewer #1: General Comments:

This manuscript describes a novel way to manipulate asymmetry via changing speeds on a treadmill within a stride. There were two main experiments, one where the speed changed based on the individuals propulsive force, and another where the speed changed during different parts of a gait cycle and a metronome was used to help the individual walk appropriately. I think the clinical applications of this work will be a great addition to the field and this study in healthy individuals shows promising results for this particular task. My primary concerns are in the statistical analysis and the role the metronome plays in influencing walking.

Major comments:

1. Even though these are healthy adults, did you do any tests to determine if these individuals walked symmetrically or asymmetrically to begin with? It seems most of the analysis was done during the period where the treadmill changed. I think the study would benefit from including a comparison to baseline walking, to see what changes the system is actually producing compared to how they typically walk. If all the analyses are within a condition comparing right and left, they do not test if that is how someone always walked or if the change in treadmill speed caused the asymmetry change.

2. While I agree that the metronome was a good way to get individuals to alter their walking pattern so the speeds could change at different portions of the gait cycle, I am curious if the authors think the metronome has any negative effects on the participant’s gait. For example, research has shown that walking to a metronome can alter individual’s gait variability. Therefore, are some of the changes in results in experiment 2 due to the metronome changing their natural gait or solely due to the treadmill speed?

3. Why did you choose a paradigm where the belts changed between 2 speeds instead of looking at an adaptive speed treadmill? Are there particular benefits of your approach versus the adaptive speed treadmill literature? While I think your approach is novel and has a lot of benefit, it may be worthwhile to further elaborate on the benefit of this versus the adaptive speed treadmill since there has been a variety of studies recently focused on that. This may be worth to touch on in the discussion.

4. I think it would be of more benefit to readers and help the reproducibility of the experiment if the statistics were presented in the text and in figures/tables for the actual manuscript and not just in the supplementary tables and figures.

5. Could you alter your presentation of your results to better match your statistical tests. For example, in the table, it seems that time may not be significant for many variables, but most of the text just emphasizes left versus right. I think it would be beneficial to the reader if both significant and insignificant results were presented.

6. Please reorganize your statistical tables. Based on what you had written in your methods, it seems you ran a separate ANOVA for each dependent variable (step length, step time, etc). However the way you present them by grouping by factor is confusing. I would have a table for each statistical test run and present it in the results.

Minor comments:

7. I would suggest presenting more of your statistical results (p values, etc) in the abstract.

8. Lines 63-65 state that: “Moreover, we expected that this would drive predictable, asymmetric changes in kinematic, kinetic, and spatiotemporal gait parameters that would depend on the timing of the speed change within the gait cycle.” Based on this hypothesis, it seems that the different timing of the speed changes would be compared to see if they presented differences. However, it did not seem the statistical analysis were set up that way. I would suggest to better align the hypotheses with the statistical approach.

9. How was the sample size determined? Did you do an apriori power analysis?

10. Line 104 mentions the treadmill acceleration was set to 6.0 m/s2. Did you consider looking at any mediolateral gait variables? I am curious if the fast acceleration influences stability. I think this would be important, especially for the clinical population application.

11. To clarify, for experiment 2, did the speeds change always at the same frequency for all four conditions, but it was just when the metronome occurred (signifying when to step) that changed?

12. For the statistical analysis, is there a certain reason why asymmetry or an asymmetry index was not used, and instead right vs left was compared? Calculating asymmetry may simplify some of the results and could directly target how asymmetry changed from normal walking to the controller conditions.

13. In the results, the time series plots are referenced when discussing the results of the statistical tests. I assume the first 30 or last 30 (the values from the bar plots) were used for the statistics. Therefore, I would suggest to update the reference figure to the bar plot or the statistical tables.

14. Did you have some way of seeing if individuals kept with the metronome?

15. For the beginning of each results section under experiment 2, it seems that some information would be better suited for the methods. For example in lines 236-240: “The Slow trial was designed to be the inverse of the Fast trial – here, participants walked with the open-loop controller activated and the treadmill moved slow over the first and last ~25% of the gait cycle and fast over the middle ~50% of the gait cycle (Figure 5A) – to test the robustness of our findings in the Fast trial (i.e., we expected the results to be opposite those observed in the Fast trial).” This describes what was done in the trial and doesn’t give any specific results. Therefore I would suggest to move these explanations to the methods.

Reviewer #2: Review of PONE-D-22-20229

General Comments

The authors examined, through two different experiments, how changes in treadmill belt velocity drive adaptations to tied-belt treadmill walking. In general, they found that their closed-loop controller drove significant and limb-specific changes in propulsive and braking forces, while the open-loop controller using a metronome did not reveal significant changes in these outcomes. This paper works to address the need to fine-tune treadmill rehabilitation protocols to better understand how specific exposures (time, perturbation magnitude, etc.) relate to altered gait outcomes. This is a well-written paper and both experiments are cleverly designed. However, I am relatively unconvinced that the authors’ observed outcomes are predictable and clinically meaningful. Below are specific comments that elaborate on this point.

Specific Comments

-Abstract, Lines 6-11: These sentences are a bit confusing. Don’t both legs generate propulsive forces during push off, regardless of if they are working to accelerate or decelerate the body?

-Introduction, Lines 59-65: In their current state, the hypotheses are not phrased in a very testable way. For example, how exactly would the proposed asymmetric walking environment drive predictable changes? In which direction would these changes occur? Predictable according to whom? As an aside, consider how the word predictable is being used; in my mind (and perhaps some readers) this implies a regression/machine learning type of approach.

-Methods, Participants: Why were only 10 participants per experiment recruited? Was a power analysis performed in advance? This sample size seems small relative to most split-belt treadmill walking studies.

-Methods, Lines 96-108: Based on my understanding, the closed-loop controller was designed to accelerate both belts when the right leg was generating propulsive forces, and decelerated both belts when the left leg was in propulsion. As participants were likely in single limb stance when the belts were accelerating/decelerating, this seems like a pretty large perturbation. For example, if the right leg was in the propulsive phase a person would be simultaneously accounting for belt acceleration as well as driving their limb forward. Was this perturbation challenging? Also, was the belt still accelerating when the contralateral limb made contact with the treadmill? How was 6 m/s^2 determined to be the acceleration of the belts?

-Methods, Lines 125-127: How was a GRF threshold used to calculate average stride frequency?

-Methods, Lines 133-135: How were treadmill speed change timing data synchronized with specific gait events?

-Methods, Lines 122-135: Figure 1 is very nicely done and explains the experimental procedures well. However, the way the procedure is outlined in the methods section is pretty unclear, it was quite difficult to understand the experiment 2 procedure. Consider rewriting this paragraph so that it aligns with Figure 1.

-Methods, Lines 143-146: Why was this method of gait event detection used instead of GRF? It seems that basing it on limb angle slope might not be the most accurate method, as the limb angle could stay relatively constant following heel strike (during foot flat) before rotation at the ankle occurs. At the very least, do you have a citation that you can include that has validated that method?

-Methods, Lines 148-149: How were limb angles calculated, specifically? Is this similar to a sagittal inclination angle?

-Methods, Lines 161-162: Isn’t this technically a repeated measures (or mixed model) MANOVA, rather than ANOVA? Same applies to statistical analysis for Experiment 2.

-Results: I think it would make it easier for the readership to include the statistical results in the methods section rather than supplementary materials.

-Results, Experiment 1: were there any time differences (i.e., early and late epochs)?

-Results, Figures: Can you include legends in your figures to make it easier to determine what each color on each graph represents? I do see it is included in the captions but it may make for easier readability with the addition of legends.

-Discussion, Lines 296-298: Can you give a specific example as to how this protocol could help with a rehab approach to a given clinical population? I believe this is quite important as one of the main cruxes of this paper is to design a procedure that can be replicated in a clinic.

-Discussion: In general, were your hypotheses supported? Why or why not?

-Discussion, Lines 303-304: One paper is cited that demonstrates longer-term effects of split-belt adaptation in clinical populations, and the results of that paper are not very convincing (Reisman et al). Are we sure there is a long-term effect? Since this paper is ultimately aimed at a long-term clinical intervention, it may make sense to include a bit more discussion in terms of current knowledge of long-term gait adaptation.

-Discussion, Lines 317-319: This is a very interesting point that I think needs further elaboration. What is happening, biomechanically or physiologically, that is causing these differences in propulsive impulses to occur? Also, in some ways this finding detracts from one of the main themes of the paper that adaptation is ‘predictable.’

-Discussion, Lines 325-327: Is there an alternative explanation as to why there was no observed preference in outcomes for experiment 2 (e.g., exposure, magnitude of speed changes, etc.)?

-Discussion, Lines 328-334: If these treadmills are not widely available, then what exactly would a clinician be able to do with the information from this paper? Is there a modified version of your experiment that can be replicated in the clinic?

6. PLOS authors have the option to publish the peer review history of their article (what does this mean?). If published, this will include your full peer review and any attached files.

Reviewer #1: No

Reviewer #2: No

---

## [Author Response · Author response to Decision Letter 0]

16 May 2023

We thank the reviewers and editor for the helpful comments and for the opportunity to revise our manuscript. We include point-by-point responses to each reviewer comment below. All line numbers refer to the copy of the manuscript with changes tracked.

Reviewer #1: General Comments:

This manuscript describes a novel way to manipulate asymmetry via changing speeds on a treadmill within a stride. There were two main experiments, one where the speed changed based on the individuals propulsive force, and another where the speed changed during different parts of a gait cycle and a metronome was used to help the individual walk appropriately. I think the clinical applications of this work will be a great addition to the field and this study in healthy individuals shows promising results for this particular task. My primary concerns are in the statistical analysis and the role the metronome plays in influencing walking.

We thank the reviewer for their kind words and constructive feedback throughout the review.

Major comments:

1. Even though these are healthy adults, did you do any tests to determine if these individuals walked symmetrically or asymmetrically to begin with? It seems most of the analysis was done during the period where the treadmill changed. I think the study would benefit from including a comparison to baseline walking, to see what changes the system is actually producing compared to how they typically walk. If all the analyses are within a condition comparing right and left, they do not test if that is how someone always walked or if the change in treadmill speed caused the asymmetry change.

We calculated asymmetry parameters for all gait parameters as:

asymmetry = (right parameter – left parameter)/(right parameter + left parameter)

We found that all participants were largely symmetric at baseline across all parameters (i.e., there were no significant differences when using one-sample t-tests to compare each parameter against zero asymmetry for any of the baseline trials).

We now include this information in the methods and results sections on lines 182-185, 209-211, and 247-249:

“We also performed a series of one-sample t-tests to compare the asymmetries in each gait parameter against zero to confirm that all participants walked relatively symmetrically at baseline (the same analyses were also performed in Experiment 2).”

“The participants walked relatively symmetrically at baseline, as there were no significant differences in the asymmetry values for any gait parameters when compared to zero (all p > 0.05).”

“As in Experiment 1, the participants walked relatively symmetrically at baseline with no significant differences in the asymmetry values for any gait parameters when compared to zero (all p > 0.05).”

2. While I agree that the metronome was a good way to get individuals to alter their walking pattern so the speeds could change at different portions of the gait cycle, I am curious if the authors think the metronome has any negative effects on the participant’s gait. For example, research has shown that walking to a metronome can alter individual’s gait variability. Therefore, are some of the changes in results in experiment 2 due to the metronome changing their natural gait or solely due to the treadmill speed?

The reviewer brings up an interesting point and an important potential limitation. We did not collect trials of constant speed walking with a metronome, so we are unable to isolate the independent effect of metronome on gait variability. That said, we expect that the particularly large speed differential (0.75 to 1.50 m/s) would induce greater change to gait variability than the metronome alone. Nevertheless, we have added this as a limitation in our revised manuscript on lines 388-390:

“We also did not test gait parameters beyond the mean values; it is possible that this new treadmill walking approach also affects other clinically relevant aspects of gait (e.g., gait variability).”

3. Why did you choose a paradigm where the belts changed between 2 speeds instead of looking at an adaptive speed treadmill? Are there particular benefits of your approach versus the adaptive speed treadmill literature? While I think your approach is novel and has a lot of benefit, it may be worthwhile to further elaborate on the benefit of this versus the adaptive speed treadmill since there has been a variety of studies recently focused on that. This may be worth to touch on in the discussion.

Adaptive speed treadmills undoubtedly provide many great options for unique gait training paradigms. However, adaptive speed treadmills require a closed-loop controller that responds to some aspect of the user’s kinematics/kinetics/etc. Therefore, there is a need for some type of instrumentation to provide an input into the controller. 

One of our primary goals in this study was to develop a technique that could be implemented into a single-belt, non-instrumented treadmill (i.e., a conventional treadmill that a patient could access in their home or local gym). By limiting the instrumentation needed, we hope that this approach may broaden the accessibility of a potential therapeutic intervention. Furthermore, we also aimed to target gait asymmetries in particular, as these are common in several clinical populations (e.g., stroke, cerebral palsy, lower limb amputation). We are not aware of any published literature (yet) that uses adaptive speed treadmills to target gait asymmetries, so while we find the adaptive speed treadmill literature very interesting, we would prefer to keep the introduction and discussion focused on published approaches that directly target asymmetry.

4. I think it would be of more benefit to readers and help the reproducibility of the experiment if the statistics were presented in the text and in figures/tables for the actual manuscript and not just in the supplementary tables and figures.

We removed all supplementary material except Supplementary Video 1 and now report all statistical results within the tables and figures of the manuscript.

5. Could you alter your presentation of your results to better match your statistical tests. For example, in the table, it seems that time may not be significant for many variables, but most of the text just emphasizes left versus right. I think it would be beneficial to the reader if both significant and insignificant results were presented.

Given the relatively large number of variables and the different types of comparisons being made (e.g., left vs. right, early vs. late, interaction terms, etc), we chose to focus on what we considered to be the most important findings of each analysis while also providing complete results from each analysis in the tables should the reader be interested in results not highlighted in the text. When we prepared a draft that included all significant and insignificant findings in the manuscript, we found that the results section became very dense and somewhat repetitive. In our view, the current format maintains the readability of the manuscript while also providing comprehensive reporting of the results; accordingly, we would prefer to maintain the current format.

6. Please reorganize your statistical tables. Based on what you had written in your methods, it seems you ran a separate ANOVA for each dependent variable (step length, step time, etc). However the way you present them by grouping by factor is confusing. I would have a table for each statistical test run and present it in the results.

We appreciate this comment by the reviewer. We chose to present our statistical results in this way to preserve the brevity of the text and tables. As the reviewer notes, we performed a separate ANOVA for each dependent variable in each of the different trials. Accordingly, there would be a very large number of tables to present were we to report each statistical test and results in a separate table. We also suggest that the text in the results section would become very long and repetitive were we to report the results in this way. As a result, we would prefer to keep the formatting of the statistics in the current format.

Minor comments:

7. I would suggest presenting more of your statistical results (p values, etc) in the abstract.

Given the large number of analyses performed and statistically significant results, we found it challenging to add specific statistical results to the abstract and remain under the 250-word limit without removing text that we considered to be essential to the understanding of the background, methods, and results more broadly. As a result, we would prefer to maintain the current abstract without numerical results.

8. Lines 63-65 state that: “Moreover, we expected that this would drive predictable, asymmetric changes in kinematic, kinetic, and spatiotemporal gait parameters that would depend on the timing of the speed change within the gait cycle.” Based on this hypothesis, it seems that the different timing of the speed changes would be compared to see if they presented differences. However, it did not seem the statistical analysis were set up that way. I would suggest to better align the hypotheses with the statistical approach.

We thank the reviewer for this helpful suggestion and have now calculated asymmetry metrics for each gait parameter and performed statistical analyses to compare these asymmetries across the different trials in Experiment 2. These are reported in the results (lines 297-302), in Figure 10, and in Table 7.

9. How was the sample size determined? Did you do an apriori power analysis?

During pilot testing, we observed large changes in key study variables (namely step lengths and ground reaction forces) across the metronome trials. An a priori power analysis suggested a sample size of fewer than 10 participants would be sufficient. To ensure robustness of our results (and to be similar to the sample sizes included in previous locomotor learning/locomotor control studies), we included 10 participants.

10. Line 104 mentions the treadmill acceleration was set to 6.0 m/s2. Did you consider looking at any mediolateral gait variables? I am curious if the fast acceleration influences stability. I think this would be important, especially for the clinical population application.

The reviewer brings up important considerations regarding the effects this system could have on balance control. Indeed, 6 m/s2 is a particularly high acceleration selected through pilot testing so that the 0.75 m/s differential (0.75 to 1.50 m/s) could be accommodated as rapidly as possible with minimal apparent affect on balance control. Notably, persons post-stroke (a theoretical target group for this system) would likely walk with considerably lower speeds and speed differentials (e.g., 0.5 to 0.8 m/s) that could accommodate a lower acceleration. Future testing will be sure to account for potential affects on balance when optimizing acceleration, especially within patient populations.

11. To clarify, for experiment 2, did the speeds change always at the same frequency for all four conditions, but it was just when the metronome occurred (signifying when to step) that changed?

The reviewer is correct, the treadmill controller was functionally the same, accelerating and decelerating at a constant frequency for all trials. Participants were instructed to coincide their heel-strike to four distinct portions of the speed profile. 

12. For the statistical analysis, is there a certain reason why asymmetry or an asymmetry index was not used, and instead right vs left was compared? Calculating asymmetry may simplify some of the results and could directly target how asymmetry changed from normal walking to the controller conditions.

As mentioned in our response to Comment 8 above, we think this was an excellent suggestion and compared asymmetry metrics for each gait parameter across the four trials as recommended. These data are provided in Figure 10 and statistical results in Table 7.

13. In the results, the time series plots are referenced when discussing the results of the statistical tests. I assume the first 30 or last 30 (the values from the bar plots) were used for the statistics. Therefore, I would suggest to update the reference figure to the bar plot or the statistical tables.

We have now moved the full statistical figures and tables into the manuscript and refer to these at the end of each relevant paragraph. We would prefer to keep the references to the time series data within each paragraph to help guide the reader through the different figures versus directing their attention immediately to the bar graphs.

14. Did you have some way of seeing if individuals kept with the metronome?

Although we did not record synchronous metronome and heel-strike timing, we did record and report treadmill speed data delineated by heel-strikes in Figures 5A-8A to address this question about participant adherence to the metronome timing. The left figure in each panel A shows a representative subject while the right figure shows the individual participant (light gray) and group (black) means of treadmill speed from heel-strike to heel-strike. While there certainly was some between-participant variance, each figure demonstrates that participants (on average) aligned their heel-strike with the corresponding metronome. 

15. For the beginning of each results section under experiment 2, it seems that some information would be better suited for the methods. For example in lines 236-240: “The Slow trial was designed to be the inverse of the Fast trial – here, participants walked with the open-loop controller activated and the treadmill moved slow over the first and last ~25% of the gait cycle and fast over the middle ~50% of the gait cycle (Figure 5A) – to test the robustness of our findings in the Fast trial (i.e., we expected the results to be opposite those observed in the Fast trial).” This describes what was done in the trial and doesn’t give any specific results. Therefore I would suggest to move these explanations to the methods.

We thank the reviewer for the suggestion. We have relocated the preamble from each of the Fast, Slow, Accelerate, and Decelerate results sections to the methods section to better explain our trials in the appropriate location.

The methods section now contains the following sentence (lines 141-149): 

“Briefly, the during the Fast trial, the treadmill moved fast (i.e., 1.5 m/s) over the first and last ~25% of the gait cycle (i.e., right heel-strike to right heel-strike) and slow (i.e., 0.75 m/s) over the middle ~50% of the gait cycle; during the Slow trial, the treadmill moved slow over the first and last ~25% of the gait cycle (i.e., right heel-strike to right heel-strike) and fast over the middle ~50% of the gait cycle; during the Accelerate trial, the treadmill accelerated from slow to fast during heel-strike such that it would decelerate to slow in the last ~50% of the gait cycle; during the Decelerate trial, the treadmill decelerated from fast to slow during heel-strike such that it would accelerate to fast in the last ~50% of the gait cycle.”

Reviewer #2: Review of PONE-D-22-20229

General Comments

The authors examined, through two different experiments, how changes in treadmill belt velocity drive adaptations to tied-belt treadmill walking. In general, they found that their closed-loop controller drove significant and limb-specific changes in propulsive and braking forces, while the open-loop controller using a metronome did not reveal significant changes in these outcomes. This paper works to address the need to fine-tune treadmill rehabilitation protocols to better understand how specific exposures (time, perturbation magnitude, etc.) relate to altered gait outcomes. This is a well-written paper and both experiments are cleverly designed. However, I am relatively unconvinced that the authors’ observed outcomes are predictable and clinically meaningful. Below are specific comments that elaborate on this point.

We appreciate the reviewer’s thoughtful feedback and provide point-by-point responses below.

Specific Comments

-Abstract, Lines 6-11: These sentences are a bit confusing. Don’t both legs generate propulsive forces during push off, regardless of if they are working to accelerate or decelerate the body?

We thank the reviewer for pointing out this confusing wording. During push-off, the trailing limb generates a positive anteroposterior force with the ground (i.e. propulsive) coincident with the leading limb which generates a negative (i.e., braking) anteroposterior force. Our controller used the positive component of this anteroposterior force as the logic to accelerate or decelerate. We have rewritten the sentence to more clearly describe our controller (lines 101-106):

“We developed a custom D-Flow (Motek Medical, Amsterdam, Netherlands) closed-loop treadmill controller that used anteroposterior ground reaction forces (AP GRF; 1000Hz) to control the treadmill speed. The closed-loop controller set the treadmill to move at 1.50 m/s (“fast” speed) when the right AP GRF was positive (i.e., right leg generated a propulsive force) and 0.75 m/s (“slow” speed) when the left AP GRF was positive (treadmill acceleration set to 6.0 m/s2).”

-Introduction, Lines 59-65: In their current state, the hypotheses are not phrased in a very testable way. For example, how exactly would the proposed asymmetric walking environment drive predictable changes? In which direction would these changes occur? Predictable according to whom? As an aside, consider how the word predictable is being used; in my mind (and perhaps some readers) this implies a regression/machine learning type of approach.

In taking the reviewer’s comment into consideration, we have removed the word “predictable” from our hypotheses and other appropriate text. While we can visually discern similarities between trials (i.e., Fast and Slow give similar results for opposite legs), we agree that use of the word “predictable” was potentially confusing given that we have not yet developed a customizable algorithm for prediction of limb mechanics for an individual participant and specific treadmill configuration. 

-Methods, Participants: Why were only 10 participants per experiment recruited? Was a power analysis performed in advance? This sample size seems small relative to most split-belt treadmill walking studies.

During pilot testing, we observed large changes in key study variables (namely step lengths and ground reaction forces) across the metronome trials. An a priori power analysis suggested a sample size of fewer than 10 participants would be sufficient. To ensure robustness of our results (and to be similar to the sample sizes included in previous locomotor learning/locomotor control studies), we included 10 participants.

-Methods, Lines 96-108: Based on my understanding, the closed-loop controller was designed to accelerate both belts when the right leg was generating propulsive forces, and decelerated both belts when the left leg was in propulsion. As participants were likely in single limb stance when the belts were accelerating/decelerating, this seems like a pretty large perturbation. For example, if the right leg was in the propulsive phase a person would be simultaneously accounting for belt acceleration as well as driving their limb forward. Was this perturbation challenging? Also, was the belt still accelerating when the contralateral limb made contact with the treadmill? How was 6 m/s^2 determined to be the acceleration of the belts? 

Indeed, this was a fairly large perturbation. Anecdotally, none of our participants (who were healthy, young adults) indicated difficulty with general accommodation to walking with either treadmill controller. The acceleration of the belt during push-off was a primary consideration for this study. As shown in Figure 4, participants increased peak propulsion and propulsive impulse as time progressed – a result likely due to their need to increase push-off in response to the acceleration. 

The acceleration of 6 m/s^2 represents 0.125 s (0.75 to 1.5 m/s). This was selected during pilot testing to ensure the full acceleration was completed before toe-off. 

-Methods, Lines 125-127: How was a GRF threshold used to calculate average stride frequency?

We have updated this sentence to more clearly describe our approach (lines 116-120): 

“Custom MATLAB (Mathworks, Natick, MA) software first calculated each participant’s average stride frequency across the 1.125 m/s Baseline trial by taking the average of the time differences between successive right heel-strikes (based on a 20 N right vertical ground reaction force threshold) over one minute of walking.”

-Methods, Lines 133-135: How were treadmill speed change timing data synchronized with specific gait events?

In Experiment 2, the treadmill operated independently of gait events. While the treadmill changed speeds by increasing during right leg propulsion and decreasing during left leg propulsion during Experiment 1, the treadmill consistently fluctuated between 0.75 and 1.5 m/s during Experiment 2. The participants were instructed to time their heel-strike with a metronome that chimed at four different times during that speed fluctuation. 

We have added the following explanation to help with Experiment 1 (lines 101-103):

“We developed a custom D-Flow (Motek Medical, Amsterdam, Netherlands) closed-loop treadmill controller that used anteroposterior ground reaction forces (AP GRF; 1000 Hz) to control the treadmill speed.”

-Methods, Lines 122-135: Figure 1 is very nicely done and explains the experimental procedures well. However, the way the procedure is outlined in the methods section is pretty unclear, it was quite difficult to understand the experiment 2 procedure. Consider rewriting this paragraph so that it aligns with Figure 1.

In concert with feedback from Reviewer 1, we added the following sentence to our methods (reallocated from our results) to better align with the figure and better explain Experiment 2 in-text (lines 141-149):

“Briefly, the during the Fast trial, the treadmill moved fast (i.e., 1.5 m/s) over the first and last ~25% of the gait cycle (i.e., right heel-strike to right heel-strike) and slow (i.e., 0.75 m/s) over the middle ~50% of the gait cycle; during the Slow trial, the treadmill moved slow over the first and last ~25% of the gait cycle (i.e., right heel-strike to right heel-strike) and fast over the middle ~50% of the gait cycle; during the Accelerate trial, the treadmill accelerated from slow to fast during heel-strike such that it would decelerate to slow in the last ~50% of the gait cycle; during the Decelerate trial, the treadmill decelerated from fast to slow during heel-strike such that it would accelerate to fast in the last ~50% of the gait cycle.”

-Methods, Lines 143-146: Why was this method of gait event detection used instead of GRF? It seems that basing it on limb angle slope might not be the most accurate method, as the limb angle could stay relatively constant following heel strike (during foot flat) before rotation at the ankle occurs. At the very least, do you have a citation that you can include that has validated that method?

We chose this method because it can generalize to situations where measurement of GRFs is not available. This method of detecting heel-strikes is common in studies of treadmill-based locomotor learning and has been compared to other methods of gait event detection in Zeni et al., Gait & Posture, 2008. We now provide a reference to this publication within the revised manuscript. 

-Methods, Lines 148-149: How were limb angles calculated, specifically? Is this similar to a sagittal inclination angle?

We calculate limb angle as the two-dimensional vector in the sagittal plane from the iliac crest to the second metatarsal head.

-Methods, Lines 161-162: Isn’t this technically a repeated measures (or mixed model) MANOVA, rather than ANOVA? Same applies to statistical analysis for Experiment 2.

Because we performed the analyses on each dependent variable independently, we performed a series of repeated measures ANOVAS (whereas a repeated measures MANOVA would have evaluated effects on multiple dependent variables simultaneously).

-Results: I think it would make it easier for the readership to include the statistical results in the methods section rather than supplementary materials.

We now include all statistical results in the figures and text within the manuscript.

-Results, Experiment 1: were there any time differences (i.e., early and late epochs)?

We refer the reviewer to Figure 4 regarding this question. Here, we observed leg-dependent greater peak propulsion and propulsive impulses, lesser peak braking forces, and greater leading limb angles in the late vs. early epochs. 

-Results, Figures: Can you include legends in your figures to make it easier to determine what each color on each graph represents? I do see it is included in the captions but it may make for easier readability with the addition of legends.

We added a legend to all figures as requested.

-Discussion, Lines 296-298: Can you give a specific example as to how this protocol could help with a rehab approach to a given clinical population? I believe this is quite important as one of the main cruxes of this paper is to design a procedure that can be replicated in a clinic.

Since we have not yet tested this approach in clinical populations, we are wary to propose a detailed clinical intervention protocol. However, we have now added a generalized example of a potential intervention in the discussion as suggested. We focused our considerations on Experiment 2, which we suggest would be more readily replicable in a clinic given the decreased reliance on instrumentation (e.g., force plates; lines 354-361):

“While future research into potential effects of long-term gait training with this approach are needed, we could speculate, for example, a customized protocol for an individual with stroke with unilateral hemiparesis resulting in reduced forward propulsion in one leg. In this customizable protocol, we might advise the open-loop (Experiment 2) controller Slow, which differentially increased trailing limb angle and increased propulsion (Figure 5). Still, more research is needed to understand the implications of this approach for gait training in clinical populations.”

-Discussion: In general, were your hypotheses supported? Why or why not?

We have revised our hypotheses to be written without the term predictability. Our hypotheses now focus on the asymmetry between limbs. We revised lines 317-318 to read:

“We found that, in support of our hypotheses, we could drive asymmetric changes in a variety of different kinematic, kinetic, and spatiotemporal gait parameters by accelerating or decelerating the treadmill speed at different points within the gait cycle.”

-Discussion, Lines 303-304: One paper is cited that demonstrates longer-term effects of split-belt adaptation in clinical populations, and the results of that paper are not very convincing (Reisman et al). Are we sure there is a long-term effect? Since this paper is ultimately aimed at a long-term clinical intervention, it may make sense to include a bit more discussion in terms of current knowledge of long-term gait adaptation.

We have cited additional evidence for the long-term effects of split-belt treadmill walking by adding a citation to a recent meta-analysis that supports the notion that there are lasting effects to split-belt treadmill training (Dzewaltowski et al., Neurorehabilitation and Neural Repair, 2021). 

-Discussion, Lines 317-319: This is a very interesting point that I think needs further elaboration. What is happening, biomechanically or physiologically, that is causing these differences in propulsive impulses to occur? Also, in some ways this finding detracts from one of the main themes of the paper that adaptation is ‘predictable.’

We believe that ground reaction forces were, at least in part, modulated through leading and trailing limb orientations. As an example, Figures 5 and 6 show coincident increased (i.e., more negative) trailing limb angle for the limb with increased peak propulsion and propulsive impulse. 

-Discussion, Lines 325-327: Is there an alternative explanation as to why there was no observed preference in outcomes for experiment 2 (e.g., exposure, magnitude of speed changes, etc.)?

We have added a discussion of some observations that we made in addition to some speculation for how/why certain changes did not manifest as expected (lines 388-394):

“We anecdotally observed participants responding in a few different ways that we were ultimately unable to assess statistically with this design. It appeared that some participants volitionally synchronized with their “preferred” modality for short durations (e.g. Fast) but switched to other synchronized heel-strikes either deliberately or as a result of natural variation in stride time. Still, others never seemed to synchronize for more than a few steps at a time without the metronome pacing to guide heel-strikes.”

We do consider that metabolic energy expenditure could be directing some of this apparent lack of preference. Human walkers are well known to identify the least metabolically taxing walking patterns. Future work will elicit how metabolic energy expenditure might affect selection of the walking pattern on this type of treadmill.

-Discussion, Lines 328-334: If these treadmills are not widely available, then what exactly would a clinician be able to do with the information from this paper? Is there a modified version of your experiment that can be replicated in the clinic?

While the findings are not necessarily directly applicable to clinical care, we have added the following statement to the discussion to indicate that active work includes developing a treadmill capable of being placed in clinics or homes (lines 381-382): 

“Indeed, active research efforts are underway to develop such a treadmill to facilitate improved accessibility to this approach.”

---

## [Decision Letter · Decision Letter 1]

5 Jun 2023

PONE-D-22-20229R1Simple within-stride changes in treadmill speed can drive selective changes in human gait symmetryPLOS ONE

Dear Dr. Roemmich,

Thank you for submitting your manuscript to PLOS ONE. After careful consideration, we feel that it has merit but does not fully meet PLOS ONE’s publication criteria as it currently stands. Therefore, we invite you to submit a revised version of the manuscript that addresses the points raised during the review process.

We look forward to receiving your revised manuscript.

Kind regards,

Pei-Chun Kao

Academic Editor

PLOS ONE

Journal Requirements:

Reviewers' comments:

Reviewer's Responses to Questions

**Comments to the Author**

1. If the authors have adequately addressed your comments raised in a previous round of review and you feel that this manuscript is now acceptable for publication, you may indicate that here to bypass the “Comments to the Author” section, enter your conflict of interest statement in the “Confidential to Editor” section, and submit your "Accept" recommendation.

Reviewer #2: (No Response)

2. Is the manuscript technically sound, and do the data support the conclusions?

Reviewer #2: Yes

3. Has the statistical analysis been performed appropriately and rigorously? 

Reviewer #2: Yes

4. Have the authors made all data underlying the findings in their manuscript fully available?

Reviewer #2: Yes

5. Is the manuscript presented in an intelligible fashion and written in standard English?

Reviewer #2: Yes

6. Review Comments to the Author

Reviewer #2: Thanks to the authors for taking the time to address all comments. I have one further suggestion, and that is to include the specifics of your power analysis in the participants section of the paper.

7. PLOS authors have the option to publish the peer review history of their article (what does this mean?). If published, this will include your full peer review and any attached files.

Reviewer #2: No

---

## [Author Response · Author response to Decision Letter 1]

7 Jun 2023

Reviewer #2: Thanks to the authors for taking the time to address all comments. I have one further suggestion, and that is to include the specifics of your power analysis in the participants section of the paper.

We thank the reviewer for their additional time reviewing this work. We added the details of the power analysis as suggested (lines 169-175 of the clean manuscript document).

---

## [Editor Report · Decision Letter 2]

8 Jun 2023

Simple within-stride changes in treadmill speed can drive selective changes in human gait symmetry

PONE-D-22-20229R2

Dear Dr. Roemmich,

We’re pleased to inform you that your manuscript has been judged scientifically suitable for publication and will be formally accepted for publication once it meets all outstanding technical requirements.

Kind regards,

Pei-Chun Kao

Academic Editor

PLOS ONE
---

## [Editor Report · Acceptance letter]

19 Jun 2023

PONE-D-22-20229R2 

Simple within-stride changes in treadmill speed can drive selective changes in human gait symmetry 

Dear Dr. Roemmich:

I'm pleased to inform you that your manuscript has been deemed suitable for publication in PLOS ONE. Congratulations! Your manuscript is now with our production department. 

Kind regards, 

on behalf of

Dr. Pei-Chun Kao 

Academic Editor

PLOS ONE